# RGB-to-Polarization Estimation: A New Task and Benchmark Study

**Beibei Lin**[†]    **Zifeng Yuan**[†,*]    **Tingting Chen**[†]
National University of Singapore
{beibei.lin, zyuan, tingting.c}@u.nus.edu

## Abstract

Polarization images provide rich physical information that is fundamentally absent from standard RGB images, benefiting a wide range of computer vision applications such as reflection separation and material classification. However, the acquisition of polarization images typically requires additional optical components, which increases both the cost and the complexity of the applications. To bridge this gap, we introduce a new task: RGB-to-polarization image estimation, which aims to infer polarization information directly from RGB images. In this work, we establish the first comprehensive benchmark for this task by leveraging existing polarization datasets and evaluating a diverse set of state-of-the-art deep learning models, including both restoration-oriented and generative architectures. Through extensive quantitative and qualitative analysis, our benchmark not only establishes the current performance ceiling of RGB-to-polarization estimation, but also systematically reveals the respective strengths and limitations of different model families — such as direct reconstruction versus generative synthesis, and task-specific training versus large-scale pre-training. In addition, we provide some potential directions for future research on polarization estimation. This benchmark is intended to serve as a foundational resource to facilitate the design and evaluation of future methods for polarization estimation from standard RGB inputs.

## 1   Introduction

Polarization images contain rich physical information that is not captured by standard RGB cameras, such as birefringence, surface stress, roughness, and other material properties [48]. This polarization information provides valuable visual cues that enhance a variety of computer vision tasks, such as reflection separation [32, 21, 34], material analysis tasks like classification and segmentation [48, 31, 25, 58, 52, 59], and shadow removal [60].

However, polarization images are not widely accessible in practice, since their acquisition requires specialized hardware, such as polarization cameras or standard cameras equipped with a rotating polarizer (see Figure 1). These devices are often expensive and inconvenient to operate, making polarization imaging impractical for widespread or everyday use. While polarization data remains difficult to obtain, standard RGB images are widely available. This raises a fundamental question: **Can polarization information be estimated directly from RGB inputs without relying on dedicated polarization sensors?**

In response to this question, we introduce a new task: RGB-to-polarization image estimation. As illustrated in Figure 1, the goal is to leverage neural networks to estimate polarization information directly from RGB inputs. To the best of our knowledge, this is the first work that explores sensor-free polarimetric imaging using neural networks. In this work, polarization information is represented

---

*Corresponding author.    [†]Equal contribution.

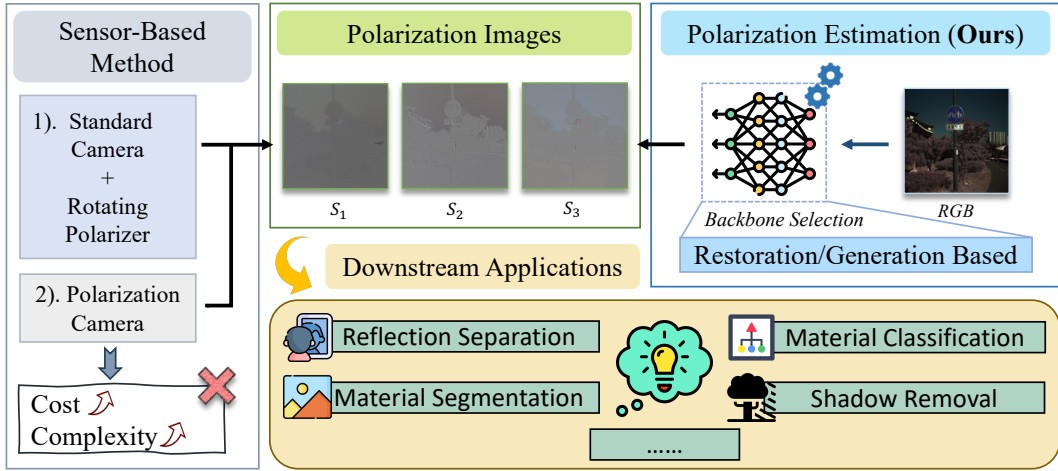

Figure 1: Comparison between sensor-based methods and our polarization estimation approach. Conventional methods rely on physical acquisition systems (e.g., polarization cameras or rotating polarizers), whereas our method leverages RGB inputs and neural networks to estimate polarization information without requiring dedicated hardware. The predicted polarization images can be readily applied to a variety of downstream tasks.

using the Stokes parameters. We treat the RGB image—corresponding to the total intensity ($S_0$)—as input, and train models to estimate the remaining polarization components, $S_1$, $S_2$, and $S_3$, which respectively characterize horizontal/vertical linear polarization (0°/90°), diagonal linear polarization ($\pm 45°$), and circular polarization. These components are essential for a wide range of downstream polarization-based vision tasks.

Building on this task, we establish a comprehensive benchmark for RGB-to-polarization image estimation. Leveraging three of the latest RGB-polarization datasets [17, 38, 19], we standardize evaluation protocols to ensure consistent and comparable results across methods. Multiple deep learning models, spanning both restoration-based and generative architectures, are further evaluated to assess their effectiveness on this task. Through systematic analysis, our benchmark reveals the strengths and limitations of existing approaches and offers insights for future development in sensor-free polarization estimation. It should be noted that multiple polarization states can correspond to the same RGB appearance, since RGB encodes only intensity and color but not the vectorial nature of light, though this ambiguity affects all methods equally and thus does not compromise the fairness of the benchmark. Our contributions are summarized as follows:

- **Task and Benchmark:** We introduce RGB-to-polarization estimation as a new computer vision task and establish the first benchmark for sensor-free polarization imaging. Built upon a recent RGB-polarization dataset, the benchmark features standardized evaluation protocols that enable consistent and reproducible comparisons across methods.

- **Comparative Study:** We evaluate a diverse set of deep learning models, including both restoration-based and generative approaches, to assess their performance on this task and uncover key limitations.

- **Insights for Future Research:** Our analysis provides practical guidance for future model design and highlights open challenges in estimating polarization information from RGB images.

## 2 Related work

**Polarization data** In the field of optics, light is typically characterized by four fundamental properties: amplitude, wavelength, phase, and polarization [3]. Among these, polarization plays a pivotal role in light–matter interactions, providing discriminative cues that help distinguish surface characteristics such as smoothness, refractive index, and coating [48].

Several mathematical models have been developed to describe the polarization state of light. The Jones vector [43, 12] describes light using two complex amplitudes for orthogonal electric field components. However, it is limited to fully polarized light and cannot account for partial or incoherent polarization, which are common in natural scenes. The Mueller matrix [20, 10] models the effect of materials on polarization, including depolarization, using a $4 \times 4$ real-valued matrix. While it accommodates both fully and partially polarized light, it demands extensive measurements and introduces 16 parameters, making it both experimentally demanding and computationally heavy. The polarimetric BRDF[37] models material-specific polarized reflectance but requires dense angular sampling. In this work, we adopt the Stokes vector representation for its ability to comprehensively describe various polarization states and its applicability to incoherent light analysis [42, 41, 30, 8]. A detailed explanation of how polarization is encoded in the Stokes parameters can be found in section 3.

**Polarization datasets**  Several polarization datasets are developed to support the analysis of polarization and spectral properties, as well as downstream vision tasks. Early works [1, 7, 26, 38] collect small-scale data using trichromatic linear-polarization cameras, typically focusing on a limited number of objects or controlled indoor scenes. More recent efforts expand to larger-scale datasets designed for specific computer vision applications, such as reflection separation [22, 29] and transparent object segmentation [31]. Fan et al. [9] present the first full-Stokes dataset that includes both linear and circular polarization components. However, the dataset consists of only 64 flat objects, which limits its applicability to real-world scenarios. Jeon et al. [17] propose a large-scale dataset that includes both trichromatic and hyperspectral Stokes measurements, covering more than 2,000 natural scenes under diverse illumination conditions. This dataset serves as the foundation for our benchmark, enabling RGB-to-polarization estimation in realistic and varied settings.

**Restoration backbones**  Existing image restoration backbones [24, 47, 56, 5, 23, 57] are widely used in low-level vision tasks and can be naturally adapted for RGB-to-polarization image estimation. Transformer-based architectures such as SwinIR [24], Uformer [47], and Restormer [56] leverage self-attention mechanisms to capture long-range dependencies, achieving strong performance across denoising, deraining, and deblurring tasks. In our benchmark, we adopt Uformer and Restormer as representative restoration backbones due to their effectiveness and generalizability.

In parallel, large-scale vision transformers pre-trained in a self-supervised manner, such as DINO [4] and MAE [14], have shown strong representation learning capabilities across a range of tasks. These models are trained on massive image collections without manual labels, and their pre-trained weights can be transferred to downstream problems with limited supervision. In our setting, we explore whether the rich semantic priors learned by MAE can facilitate RGB-to-polarization estimation. Specifically, we initialize the MAE encoder with pre-trained weights and fine-tune the entire network for RGB-to-polarization estimation.

**Generation backbones**  Existing generative models, such as GANs [11] and diffusion models [15], have demonstrated strong capabilities in modeling complex image distributions. In this work, we explore whether diffusion models can be used to estimate polarization images from RGB inputs. Specifically, we adopt two representative conditional diffusion models, WDiff [33] and DiT [36], where the RGB image guides the generation of polarization components.

In addition to training task-specific diffusion models from scratch, Stable Diffusion [40], a large-scale pre-trained model, is leveraged to estimate polarization images. Previous works [46, 27, 49] show that the priors of pre-trained diffusion models can be effectively transferred to downstream tasks. We adopt two representative models, RealFill [44] and Img2ImgTurbo [35], which are originally designed for image inpainting and translation. These models are further fine-tuned for the RGB-to-polarization estimation task, enabling the synthesis of polarization components guided by RGB context.

**Polarization-related tasks**  When extended to computer vision tasks, polarization plays a critical role in distinguishing surface properties. Specifically, the polarization state of reflected light changes depending on factors such as a material's refractive index, surface roughness, coating, or texture—enabling the classification of materials and object recognition [48]. In addition, specular reflections can be suppressed by exploiting polarization properties, and the contrast of transparent or translucent objects can be enhanced using a polarizer [39]. Furthermore, materials exhibiting birefringence or optical activity cause characteristic changes in polarization, often reflected in the $S_2$ and $S_3$ components, which are useful for biomedical imaging, tissue analysis, and fiber inspection

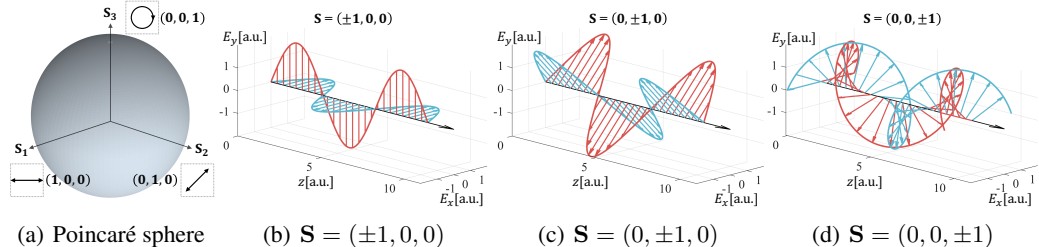

(a) Poincaré sphere    (b) $\mathbf{S} = (\pm 1, 0, 0)$    (c) $\mathbf{S} = (0, \pm 1, 0)$    (d) $\mathbf{S} = (0, 0, \pm 1)$

Figure 2: (a) The Poincaré sphere serves as a geometric representation for describing all possible states of polarization using the Stokes parameters. (b–d) Polarization describes how the electric field of a light wave oscillates within the plane perpendicular to the direction of propagation. (b) Linear polarizations at 0° and 90°. (c) Linear polarizations at ±45° angles. (d) Right- and left-handed circular polarizations.

[8]. Finally, surface normals affect the polarization state of reflected light, which in turn provides geometric cues for shape or depth inference in tasks such as Shape-from-Polarization (SfP) based 3D reconstruction [18]. Beyond SfP, polarization has also been exploited for reflectance and appearance modeling [2, 16, 13], as well as for task-specific applications including polarization-aware semantic segmentation [28], sparse polarization sensing [19], and wearable robotics [50]. These cues are absent in standard RGB images, highlighting the unique and valuable role of polarization information.

## 3    Method

In this section, we first provide a brief overview of the Stokes parameters that characterize the polarization state of the light, followed by the formulation of the RGB-to-polarization estimation task and the construction of our benchmark, including dataset preparation, evaluation protocols, and model evaluation.

### 3.1    Stokes parameters and derived quantities

**Stokes parameters**    The polarization state of light characterizes how the electric field oscillates within the plane perpendicular to the direction of propagation. In this work, we represent the polarization state of light using the Stokes vector and visualize it on the Poincaré sphere, as illustrated in Figure 2, where each point corresponds to a unique polarization state[30, 53, 42, 55, 43]. A light wave's polarization can be described using the four Stokes parameters $(S_0, S_1, S_2, S_3)$, where $S_0$ denotes the total intensity of the beam, and the remaining components collectively characterize the polarization state. Specifically, $S_1$ represents the difference in intensity between 0° and 90° linear polarization; $S_2$ corresponds to the difference between +45° and -45° linear polarization; and $S_3$ indicates the difference between right- and left-handed circular polarization [30, 42, 54].

Mathematically, the Stokes parameters can be expressed in terms of the polarization azimuth angle $\psi$ and the ellipticity angle $\chi$ as:

$$S_1 = S_0 \cos(2\psi) \cos(2\chi) \tag{1}$$
$$S_2 = S_0 \sin(2\psi) \cos(2\chi) \tag{2}$$
$$S_3 = S_0 \sin(2\chi) \tag{3}$$

where $S_0$ denotes the total light intensity, represented as an RGB image. On the Poincaré sphere, the polarization direction is represented by the vector $\mathbf{S} = (S_1, S_2, S_3)/S_0$. For instance, $\mathbf{S} = (1, 0, 0)$ corresponds to 0° linear polarization, $(-1, 0, 0)$ to 90° linear polarization, $(0, 1, 0)$ and $(0, -1, 0)$ to ±45° linear polarization, and $(0, 0, \pm 1)$ to right- or left-handed circular polarization. Elliptical polarization states lie between these extremes, depending on the value of $S_3$. Each component of the Stokes vector carries distinct material-sensitive information: high $S_1$ values often indicate smooth dielectric or metallic surfaces due to dominant specular reflection; elevated $S_2$ suggests birefringent or fibrous materials; and strong $S_3$ values arise in scattering or optically active media, such as biological tissues or rough dielectrics [39].

**Interpretable polarization features**   While the Stokes parameters $(S_0, S_1, S_2, S_3)$ provide a complete physical description of polarization, they are not always the most interpretable for human observers or visual analysis. To better convey polarization properties, several derived representations are commonly used. One of the most fundamental is the degree of polarization (DoP), which quantifies the fraction of total intensity carried by the polarized component[45]:

$$\text{DoP} = \frac{\sqrt{S_1^2 + S_2^2 + S_3^2}}{S_0} = \sqrt{\left(\frac{\sqrt{S_1^2 + S_2^2}}{S_0}\right)^2 + \left(\frac{S_3}{S_0}\right)^2} = \sqrt{\text{DoLP}^2 + \text{CoP}^2} \qquad (4)$$

This unified expression relates DoP to its two physically meaningful components: the degree of linear polarization (DoLP) and the circular polarization ratio (CoP), defined respectively as:

$$\text{DoLP} = \frac{\sqrt{S_1^2 + S_2^2}}{S_0}, \quad \text{CoP} = \frac{S_3}{S_0} \qquad (5)$$

DoLP characterizes the proportion of linearly polarized light, while CoP indicates the contribution of circular polarization. In addition to magnitude, the angle of linear polarization (AoLP) describes the orientation of linear polarization and is defined as:

$$\text{AoLP} = \frac{1}{2}\arctan\left(\frac{S_2}{S_1}\right) \qquad (6)$$

This formulation yields angles in the range $[-90°, 90°]$, representing the azimuthal angle of the polarization ellipse.

Together, these derived features—DoP, DoLP, CoP, and AoLP—offer more interpretable and visually meaningful descriptions of polarization than raw Stokes components. They also exhibit distinct gradient distributions: for example, AoLP often presents sharper local variations due to angular wrapping, whereas DoLP and CoP follow hyper-Laplacian-like distributions[17]. This highlights the need for feature-specific priors and visualization strategies when analyzing polarization images.

## 3.2   Benchmark design

**Task definition**   We define RGB-to-polarization image estimation as a pixel-wise prediction task, where the goal is to estimate polarization information from a single RGB input image. Given an RGB image $\mathbf{I}_{\text{RGB}} \in \mathbb{R}^{H \times W \times 3}$, where $H \times W$ denotes the spatial resolution, the objective is to estimate Stokes components $\mathbf{S} \in \mathbb{R}^{H \times W \times 9}$. Each Stokes component, $S_1$, $S_2$, and $S_3$, is represented as a 3-channel image. The final output is obtained by concatenating these components along the channel dimension, resulting in $\mathbf{S} = [S_1, S_2, S_3]$.

**Dataset**   Next, we build our benchmark on the recent large-scale RGB-polarization dataset proposed by Jeon et al. [17]. It provides high-quality, spatially aligned image pairs consisting of RGB inputs $S_0$ and their corresponding Stokes components $[S_1, S_2, S_3]$, enabling supervised training. All images are resized to a fixed resolution of $H \times W$ before training. Both the RGB inputs and the Stokes targets are normalized to the range $[0, 1]$ for consistent training across models.

**Evaluation metrics**   Following standard practices in low-level vision, we evaluate model performance using three complementary metrics: peak signal-to-noise ratio (PSNR), structural similarity index (SSIM), and learned perceptual image patch similarity (LPIPS). PSNR and SSIM assess pixel-level accuracy and structural fidelity, while LPIPS captures perceptual similarity in the feature space. All metrics are computed independently for each Stokes component and then averaged to report the overall performance. These metrics also align with the physical meaning of the Stokes components: high PSNR and SSIM and low LPIPS indicate that the predicted Stokes maps better preserve the spatial and structural polarization cues of the ground truth.

## 3.3   Baselines

With the defined benchmark, we evaluate a range of representative deep learning models that fall into two categories: restoration-based and generation-based approaches.

Table 1: Quantitative results on RGB-to-polarization image estimation. We report PSNR, SSIM, and LPIPS for each estimated Stokes component ($S_1$, $S_2$, $S_3$), as well as their averages. Higher PSNR and SSIM and lower LPIPS indicate better performance. We highlight the best and second-best results for each component and the averages.

| Method | $S_1$ | | | $S_2$ | | | $S_3$ | | | Average | | |
|---|---|---|---|---|---|---|---|---|---|---|---|---|
| | PSNR | SSIM | LPIPS | PSNR | SSIM | LPIPS | PSNR | SSIM | LPIPS | PSNR | SSIM | LPIPS |
| Wdiff [33] | 10.62 | 0.6454 | 0.4617 | 14.65 | 0.7172 | 0.2817 | 14.05 | 0.6840 | 0.3882 | 13.11 | 0.6822 | 0.3772 |
| DiT [36] | 20.96 | 0.7984 | 0.3238 | 24.01 | 0.8607 | 0.1680 | 25.02 | 0.8636 | 0.2424 | 23.33 | 0.8409 | 0.2447 |
| Realfill [44] | 20.43 | 0.8194 | 0.2964 | 21.92 | 0.7573 | 0.2292 | 23.07 | 0.8308 | 0.2707 | 21.81 | 0.8025 | 0.2654 |
| Img2ImgTurbo [35] | 21.47 | 0.8532 | 0.3931 | 23.65 | 0.8522 | 0.3465 | 24.87 | 0.9122 | 0.3230 | 23.33 | 0.8725 | 0.3542 |
| Restormer [56] | 22.54 | 0.8495 | 0.3072 | 24.42 | 0.8764 | 0.1693 | 24.99 | 0.8932 | 0.2341 | 23.98 | 0.8730 | 0.2369 |
| Uformer [47] | 22.61 | 0.8482 | 0.3007 | 24.72 | 0.8721 | 0.1624 | 25.69 | 0.8963 | 0.2170 | 24.34 | 0.8722 | 0.2267 |
| MAE [14] | 22.73 | 0.8690 | 0.3521 | 25.54 | 0.8759 | 0.2194 | 25.94 | 0.9179 | 0.2338 | 24.74 | 0.8876 | 0.2684 |

**Restoration-based approaches**    Two representative restoration backbones, Restormer [56] and Uformer [47], are selected for evaluation. Both models are originally designed for image-to-image restoration tasks, where the input and output are 3-channel RGB images. They have demonstrated strong performance on tasks such as denoising, deraining, and deblurring, making them suitable baselines for pixel-wise polarization prediction. In our implementation, we modify the output layer to produce 9 channels, corresponding to the concatenated Stokes components $\mathbf{S} \in \mathbb{R}^{H \times W \times 9}$ for supervision. The models are trained using an $L_1$ loss between the predicted and ground-truth Stokes components.

Then we further evaluate Masked Autoencoder (MAE) [14], a vision transformer pre-trained using self-supervised learning on large-scale image datasets. MAE learns strong visual representations by reconstructing randomly masked image patches. With the pre-trained parameters, it can extract robust features for a variety of downstream tasks. In our setting, we modify the output channels of the final predictor projection layer to 9, corresponding to the concatenated Stokes components $\mathbf{S} \in \mathbb{R}^{H \times W \times 9}$ used for supervision. Both the MAE encoder and decoder are initialized with pre-trained weights. The predictor projection layer is initialized by inflating the original parameters through channel-wise replication to match the 9-channel output. The entire model is then fine-tuned end-to-end using an $L_1$ loss between the predicted and ground-truth Stokes components.

**Generation-based approaches**    Next, we explore generative diffusion models for RGB-to-polarization estimation. Two types of models are evaluated: (1) task-specific diffusion models trained from scratch, and (2) large-scale pre-trained models adapted to our task.

For the first type, we adopt WDiff [33] and DiT [36], two representative conditional diffusion models. Both are trained to iteratively denoise a latent polarization representation conditioned on the input RGB image. In our setting, we treat the RGB image as the conditioning input and train the diffusion process to generate the 9-channel Stokes output $\mathbf{S} \in \mathbb{R}^{H \times W \times 9}$. The training objective minimizes the denoising error over the diffusion steps using standard diffusion loss formulations.

For the second type, we adapt pre-trained diffusion models to the polarization estimation task. Specifically, we fine-tune RealFill [44] and Img2ImgTurbo [35], which are originally developed for image inpainting and translation. These models are built upon Stable Diffusion [40] and leverage rich visual priors learned from large-scale image-text datasets. Instead of modifying the model architecture, we train a separate model for each Stokes component, i.e., $S_1$, $S_2$, and $S_3$, where each component is represented as a 3-channel image. This allows us to directly reuse the pre-trained UNet decoder without altering the output layer. Each model is fine-tuned in a conditional generation setting, where the RGB image serves as guidance. During training, we follow the default configurations and hyperparameters provided in the original implementations.

# 4    Experiments

In our experiments, we primarily adopt the dataset provided by Jeon et al. [17] as the training and evaluation benchmark. Specifically, the first 1,000 RGB–Stokes image pairs are used for training, and the last 200 pairs from the dataset are reserved for testing. For a fair comparison, all image pairs are

Table 2: Quantitative results on RGB-to-polarization image estimation across datasets. We report average PSNR, SSIM, and LPIPS over Stokes components on Jeon et al. [17], Qiu et al. [38], and Kurita et al. [19] datasets. Higher PSNR/SSIM and lower LPIPS indicate better performance. We highlight the best and second-best results for each column.

| Method | Jeon et al. [17] | | | Qiu et al. [38] | | | Kurita et al. [19] | | |
|---|---|---|---|---|---|---|---|---|---|
| | PSNR | SSIM | LPIPS | PSNR | SSIM | LPIPS | PSNR | SSIM | LPIPS |
| Wdiff [33] | 13.11 | 0.6822 | 0.3772 | 11.44 | 0.6523 | 0.4042 | 11.56 | 0.6945 | 0.4311 |
| DiT [36] | 23.33 | 0.8409 | 0.2447 | 14.74 | 0.7328 | 0.2829 | 17.86 | 0.8191 | 0.2874 |
| RealFill [44] | 21.81 | 0.8025 | 0.2654 | 15.19 | 0.7241 | 0.3028 | 18.09 | 0.8252 | 0.2654 |
| Img2ImgTurbo [35] | 23.33 | 0.8725 | 0.3542 | 15.65 | 0.7566 | 0.3811 | 18.78 | 0.8504 | 0.3115 |
| Restormer [56] | 23.98 | 0.8730 | 0.2369 | 14.77 | 0.5325 | 0.5186 | 18.85 | 0.8394 | 0.2697 |
| Uformer [47] | 24.34 | 0.8722 | 0.2267 | 14.68 | 0.7278 | 0.3160 | 18.74 | 0.8381 | 0.2627 |
| MAE [14] | 24.74 | 0.8876 | 0.2684 | 15.02 | 0.7401 | 0.3238 | 18.81 | 0.8499 | 0.2772 |

resized to 256×256 for training and testing across all baseline models. In addition to the in-dataset evaluation, we further assess the generalization ability of the trained models on two external datasets: Qiu et al. [38] and Kurita et al. [19].

## 4.1 Implementation details

All experiments are conducted on a server equipped with four NVIDIA RTX A5000 GPUs, each with 24 GB of memory. To ensure reproducibility, we will release all code, model checkpoints, and configuration files upon acceptance. The following paragraphs describe the network architectures of all evaluated models, while detailed training hyperparameters are deferred to the Appendix.

**Restoration baselines** Restormer [56] consists of four hierarchical levels, each containing a series of transformer blocks. The numbers of transformer blocks in the four levels are set to 4, 6, 6, and 8, respectively. The embedding dimension is set to 48. Uformer [47] follows an encoder–decoder architecture with four encoder layers and four decoder layers. Each layer contains two transformer blocks, and the embedding dimension is set to 32. MAE [14] adopts an asymmetric encoder–decoder design, where the encoder and decoder contain 24 and 8 transformer blocks, respectively.

**Generation baselines** WDiff [33] adopts a hierarchical U-Net architecture with six resolution levels, each containing two residual blocks. Attention is applied at the $16 \times 16$ resolution. DiT [36] is a Vision Transformer-based diffusion model with 10 transformer blocks, each using 6 attention heads and a hidden size of 768. RealFill [44] is built on top of Stable Diffusion, using a U-Net backbone with cross-attention. It incorporates Low-Rank Adaptation (LoRA) modules with rank 8 and dropout 0.1 to fine-tune both the UNet and the text encoder. Only LoRA parameters are updated during training, while the base model remains frozen. Img2ImgTurbo [35] also builds on Stable Diffusion, but improves generation efficiency by directly injecting the RGB latent into the denoising UNet. It integrates LoRA adapters into both the UNet and VAE components, with LoRA ranks set to 8 and 4, respectively, enabling lightweight and flexible fine-tuning.

## 4.2 Quantitative evaluation

Table 1 shows the quantitative results for RGB-to-polarization estimation across all baseline models. For each method, we evaluate the predicted Stokes components ($S_1$, $S_2$, and $S_3$) using three commonly used metrics: PSNR, SSIM, and LPIPS. Among all methods, MAE [14] achieves the best overall performance, attaining the highest average PSNR (24.74), SSIM (0.8876), and a competitive LPIPS (0.2684). While Restormer [56] and Uformer [47] are trained from scratch, both models also exhibit strong performance, particularly in structural fidelity and perceptual similarity. Notably, Uformer achieves the lowest average LPIPS score of 0.2267 across all evaluated methods.

Diffusion-based models exhibit inconsistent performance across different architectures. WDiff [33] shows relatively low reconstruction quality, while DiT [36] demonstrates notable improvements across all metrics. Among the pre-trained diffusion models, Img2ImgTurbo [35] consistently outperforms RealFill [44] on most metrics, particularly in PSNR and SSIM. These results establish a

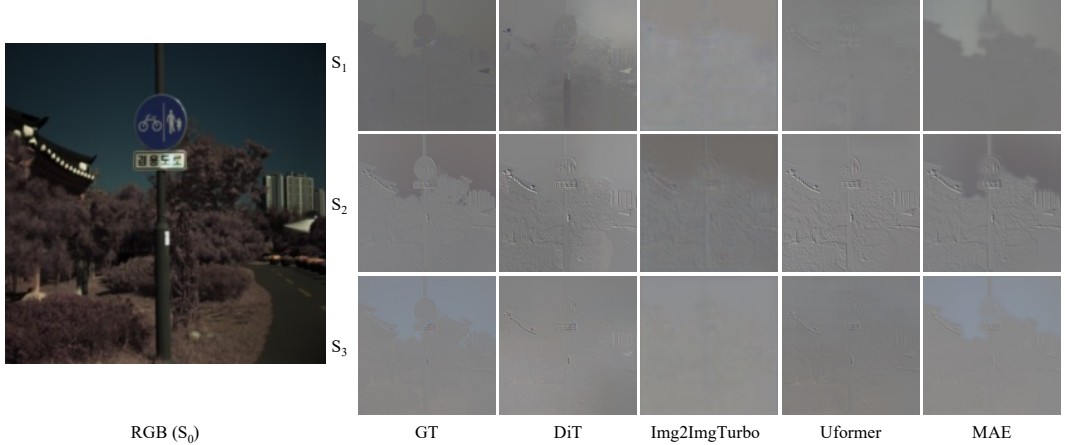

RGB ($S_0$)  GT  DiT  Img2ImgTurbo  Uformer  MAE

Figure 3: Qualitative comparison of estimated polarization components from RGB input. Results are shown for Uformer [47], MAE [14], DiT [36], and Img2ImgTurbo [35].

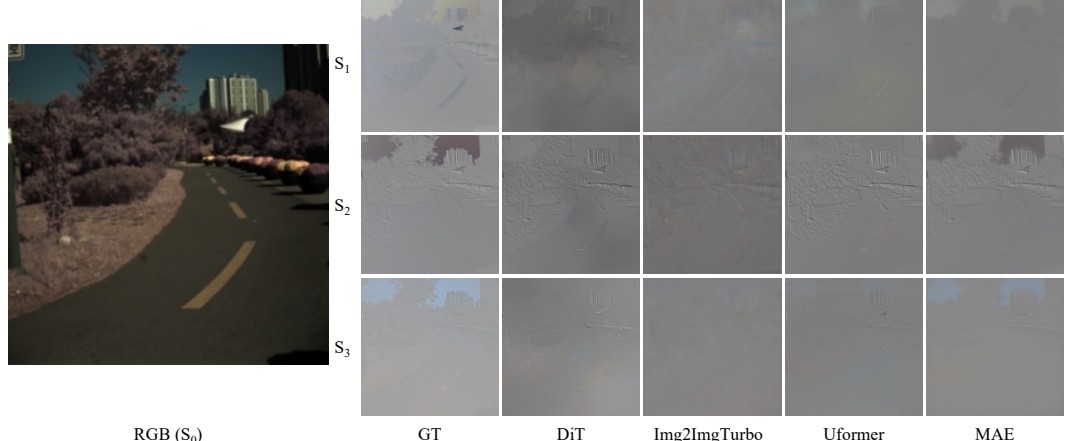

RGB ($S_0$)  GT  DiT  Img2ImgTurbo  Uformer  MAE

Figure 4: Qualitative comparison of estimated polarization components from RGB input. Results are shown for Uformer [47], MAE [14], DiT [36], and Img2ImgTurbo [35].

clear performance landscape across different model families and provide a foundation for further comparative analysis.

These quantitative trends can also be understood from the perspective of physical polarization properties. We find that estimating the $S_1$ component is more difficult than $S_2$ and $S_3$. Physically, $S_1$ represents the difference between horizontal and vertical polarization, which tends to be more sensitive to surface orientation and material properties. In many natural scenes, this component is weaker or more spatially uniform due to diffuse reflection, leading to a lower signal-to-noise ratio and making it harder for the model to learn accurate patterns from RGB inputs.

To further assess the generalization ability of RGB-to-polarization estimation, we evaluate models trained on the dataset provided by Jeon et al. [17] using two additional benchmarks: Qiu et al. [38] and Kurita et al. [19]. As shown in Table 2, the results exhibit consistent trends across datasets: restoration-based and pre-trained models (e.g., MAE, Uformer) generally achieve stronger performance, while diffusion-based models lag behind in quantitative metrics.

## 4.3 Qualitative evaluation

Figures 3 and 4 show qualitative results of the estimated Stokes components generated by DiT [36], Img2ImgTurbo [35], Uformer [47], and MAE [14]. Each column visualizes the reconstructed

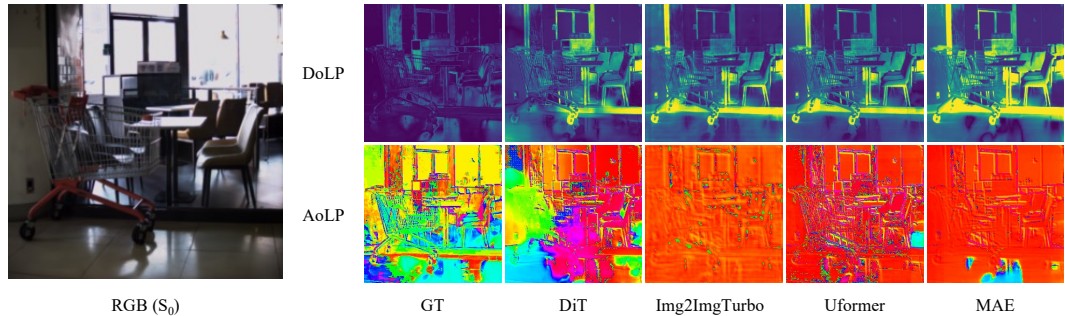

Figure 5: Qualitative comparison of predicted DoLP and AoLP maps, derived from Stokes components estimated from RGB input. Ground-truth maps are provided for reference, along with results from Uformer [47], MAE [14], DiT [36], and Img2ImgTurbo [35].

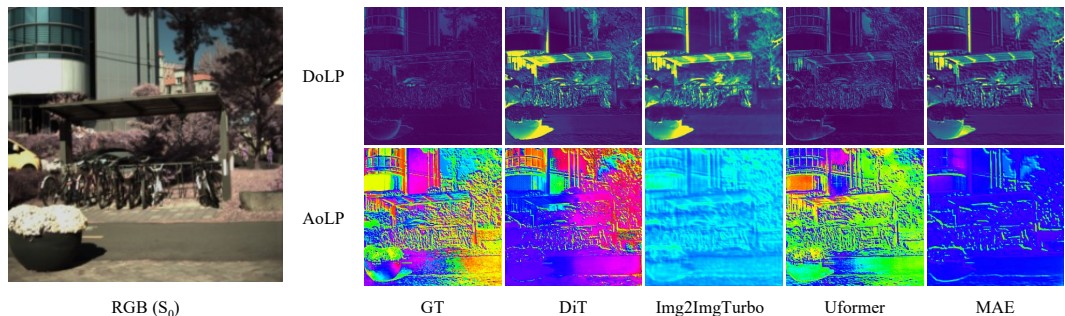

Figure 6: Qualitative comparison of predicted DoLP and AoLP maps, derived from Stokes components estimated from RGB input. Ground-truth maps are provided for reference, along with results from Uformer [47], MAE [14], DiT [36], and Img2ImgTurbo [35].

$S_1$, $S_2$, and $S_3$ components produced by each method. The results reveal diverse reconstruction characteristics across models, including differences in texture sharpness, structural consistency, and polarization pattern styles. Overall, MAE and Uformer demonstrate superior visual quality compared to diffusion-based approaches. MAE is particularly effective in preserving fine structural details and material boundaries, with minimal artifacts across all Stokes components. However, the restored details from restoration-based methods still deviate from the correct results, indicating that such models may still lack the capacity to capture the underlying polarization cues precisely. In contrast, diffusion-based methods struggle to reconstruct the correct appearance and structure of polarization components, indicating difficulty in modeling these signals. Although Img2ImgTurbo achieves higher PSNR scores than DiT, its visual results are less faithful to the ground truth, particularly in terms of structural detail. This suggests a trade-off between pixel-level accuracy and perceptual quality among diffusion-based models.

Furthermore, Figures 5 and 6 visualize the corresponding DoLP and AoLP maps derived from the estimated Stokes components. These results provide an alternative perspective by directly reflecting the physical polarization cues. The restoration-based Uformer generally produces more accurate and stable DoLP patterns, whereas the diffusion-based DiT achieves relatively better AoLP maps, highlighting their complementary strengths in capturing polarization information.

These qualitative observations highlight that both restoration-based and generative-based methods have limitations in RGB-to-polarization estimation. Restoration models may lack fine polarization accuracy, while generative models often struggle with structural consistency.

### 4.4 Discussion

**Restoration vs. Generation**   Table 1 reveals a clear performance gap between restoration-based and generation-based approaches. Restoration models, such as Restormer and Uformer, consistently achieve higher PSNR and SSIM scores than diffusion-based models like WDiff and DiT. Among

the pre-trained baselines, MAE also outperforms both Img2ImgTurbo and RealFill by a substantial margin across most metrics. These results suggest that the restoration backbones are more effective for Stokes component prediction, likely due to their capacity for precise pixel-level estimation.

Although WDiff struggles to produce accurate reconstructions, more advanced diffusion models such as DiT demonstrate marked improvements across all metrics. This indicates that task-specific diffusion models still hold promise for polarization estimation, particularly with further architectural or training enhancements.

**Pre-training vs. Training from scratch**  Pre-trained models exhibit clear advantages in RGB-to-polarization estimation. MAE achieves the highest average PSNR and SSIM, outperforming Uformer by a noticeable margin, demonstrating the effectiveness of transferring rich visual representations learned from large-scale RGB data. Img2ImgTurbo also outperforms task-specific diffusion models like WDiff and DiT by leveraging the strong generative prior of Stable Diffusion. In contrast, RealFill fine-tunes only the LoRA modules in the UNet while keeping the VAE frozen. This leads to information loss during latent encoding and decoding, which degrades the quality of polarization prediction.

One contributing factor to the improved performance is the intrinsic correlation between RGB images and polarization information. Since RGB images encode overall light intensity and structural patterns, models pre-trained on RGB data are better equipped to extract cues relevant for polarization estimation, even in the absence of direct supervision. These findings indicate that incorporating pre-trained weights, whether through self-supervised representation learning or large-scale generative modeling, is a promising strategy to enhance the accuracy and robustness of sensor-free polarization prediction.

**Insights for future research**  Our benchmark reveals several insights for future research. First, although both restoration-based and generation-based methods achieve promising results, there remains significant room for improvement in fine-grained detail reconstruction and polarization fidelity. Second, our benchmark leverages existing state-of-the-art backbones to establish strong baselines. However, future research may benefit from incorporating physical constraints, as the Stokes components inherently encode rich physical properties of light. Third, since acquiring large-scale paired data for RGB-to-polarization training is challenging, future work may explore self-supervised methods using unlabeled polarization data. Fourth, although our benchmark adapts image pre-trained models for polarization estimation, the adaptation is not specifically tailored to this task. Future research may explore more effective adaptation or fine-tuning methods to better capture polarization-related features. Finally, as the estimated polarization information may contain inaccuracies, it can affect downstream applications. Estimating polarization along with a confidence map is therefore an important direction for future research.

## 5 Conclusion

This paper introduces and benchmarks a new task: RGB-to-polarization image estimation, which aims to infer polarization information from standard RGB inputs without requiring specialized sensors. We formalize the task using Stokes parameters and construct the first comprehensive benchmark based on a recent large-scale dataset. Our evaluation covers various deep learning models, including restoration-based backbones and generation-based backbones. Through extensive quantitative and qualitative analysis, we reveal the strengths and limitations of existing approaches, providing a detailed performance landscape for this underexplored problem. Our results indicate that pre-trained models, such as MAE and Stable Diffusion with LoRA, offer strong prior knowledge that can be effectively transferred to polarization estimation, leading to consistently improved performance. We hope this benchmark will serve as a foundation for future research, fostering the development of accurate and efficient polarization estimation methods from RGB images alone.

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

# Appendix

## A    Implementation details

**Training details of restoration baselines**    The training steps for Restormer [56], Uformer [47], and MAE [14] are set to 70,000, 70,000, and 40,000, respectively. Due to differences in patch size and network depth, we adopt different batch sizes for training. For each model, we use the maximum batch size that fits within GPU memory constraints, which are 16, 48, and 128, respectively. During training, we use the Adam optimizer with an initial learning rate of $1.5 \times 10^{-4}$.

**Training details of generation baselines**    The training steps for WDiff, DiT, RealFill, and Img2ImgTurbo are set to 90,000, 90,000, 10,000, and 8,000, respectively. We use the Adam optimizer with an initial learning rate of $1.5 \times 10^{-4}$ for WDiff and DiT, and adopt batch sizes of 128 and 258, respectively. For RealFill and Img2ImgTurbo, due to higher memory consumption, the batch sizes are limited to 12 and 4, respectively. To simulate a larger effective batch size, we apply gradient accumulation with a factor of 4 for Img2ImgTurbo. Following [44], we adopt separate learning rates for LoRA modules in RealFill: 2e-4 for the UNet and 4e-5 for the text encoder. For Img2ImgTurbo [35], we set a unified learning rate of 5e-6 for all learnable parameters.

## B    More experiments

### B.1    More qualitative results

We present supplementary qualitative results of the estimated Stokes components generated by DiT [36], Img2ImgTurbo [35], Uformer [47], and MAE [14]. The results are illustrated in Figures 7–23. Among the methods, MAE and Uformer produce consistently better visual quality than the diffusion-based approaches. Nonetheless, there remains substantial room for improvement in accurately estimating polarization data.

Figures 24–30 present additional results, visualizing the corresponding DoLP and AoLP maps derived from the estimated Stokes components. Similar to the main paper, these results provide a physical perspective by directly reflecting polarization cues. While the restoration-based Uformer continues to yield more accurate and stable DoLP patterns, and the diffusion-based DiT demonstrates relatively better AoLP predictions, the overall performance of all models still leaves room for improvement, particularly in handling challenging lighting conditions and fine-grained polarization details.

### B.2    Downstream task evaluation

To further investigate the practical utility of estimated polarization images, we conduct preliminary experiments on a downstream task of polarization-aware semantic segmentation. Following Liu et al. [28], we evaluate segmentation performance by replacing ground-truth polarization inputs with estimated polarization channels. Table 3 summarizes the results. We observe that performance degrades when replacing real polarization with estimated counterparts, especially when all four channels are substituted (mIoU drops from 92.45 to 45.56). However, the relatively smaller gap when substituting only one or two channels indicates that estimated polarization still provides useful cues for segmentation. This highlights both the current limitations and the potential of RGB-to-polarization estimation for downstream vision tasks.

Table 3: Segmentation performance (mIoU) on UPLight dataset using ShareCMP when replacing real polarization inputs with estimated ones.

| Inputs | mIoU (%) $\uparrow$ |
|---|---|
| $I_0, I_{45}, I_{90}, I_{135}$ (all real) | 92.45 |
| Estimated $I_0$, others real | 76.91 |
| Estimated $I_{45}$, others real | 92.40 |
| Estimated $I_{90}$, others real | 57.40 |
| Estimated $I_{135}$, others real | 75.68 |
| All estimated | 45.56 |

Table 4: Model complexity comparison including FLOPs, number of parameters, runtime, and memory usage. Runtime is measured on a single NVIDIA RTX A5000 GPU. FLOPs and parameter counts are taken from the original papers.

| Method | FLOPs | Parameters | Runtime (s) | Training Memory | Test Memory |
|---|---|---|---|---|---|
| WDiff [33] | 16 GFLOPs | 109.7M | 0.25 | $4 \times 24$G GPUs | ∼1G |
| DiT [36] | 19 GFLOPs | 108.7M | 0.19 | $4 \times 24$G GPUs | ∼1G |
| RealFill [44] | 86 GFLOPs | 1.7M | 5.00 | $4 \times 24$G GPUs | ∼3G |
| Img2ImgTurbo [35] | 86 GFLOPs | 9.5M | 0.20 | $4 \times 24$G GPUs | ∼6G |
| Restormer [56] | 38 GFLOPs | 26.1M | 0.04 | $2 \times 24$G GPUs | ∼1G |
| Uformer [47] | 11 GFLOPs | 5.3M | 0.01 | $2 \times 24$G GPUs | ∼1G |
| MAE [14] | 64 GFLOPs | 330.0M | 0.01 | $4 \times 24$G GPUs | ∼2G |

Table 5: Stability at larger resolutions. Quantitative results at both 256×256 and 512×512 input sizes are reported.

| Method | Input size: 256×256 | | | Input size: 512×512 | | |
|---|---|---|---|---|---|---|
| | PSNR | SSIM | LPIPS | PSNR | SSIM | LPIPS |
| WDiff [33] | 13.11 | 0.6822 | 0.3772 | 12.53 | 0.6765 | 0.3997 |
| DiT [36] | 23.33 | 0.8409 | 0.2447 | 18.65 | 0.8303 | 0.2394 |
| RealFill [44] | 21.81 | 0.8025 | 0.2654 | 23.74 | 0.8238 | 0.2349 |
| Img2ImgTurbo [35] | 23.33 | 0.8725 | 0.3542 | 22.52 | 0.8372 | 0.3602 |
| Restormer [56] | 23.98 | 0.8730 | 0.2369 | 23.35 | 0.8757 | 0.2379 |
| Uformer [47] | 24.34 | 0.8722 | 0.2267 | 24.11 | 0.8883 | 0.2092 |
| MAE [14] | 24.74 | 0.8876 | 0.2684 | 23.63 | 0.8650 | 0.2810 |

## B.3 Depth estimation frameworks for polarization

To investigate whether depth estimation frameworks can be applied to polarization estimation, we conducted experiments using Depth Anything v2 [51]. Specifically, we modified the last layer to output 9 channels for predicting $S_1$, $S_2$, and $S_3$, while keeping the pre-trained parameters for the remaining layers. The averaged PSNR, SSIM, and LPIPS are 25.75, 0.8777, and 0.3765, respectively.

Although this model achieves higher pixel-level metrics compared to MAE, the perceptual quality is noticeably worse. This suggests that pre-trained depth-based models provide strong structural priors and achieve better pixel-level metrics. However, their outputs tend to be over-smoothed, leading to worse perceptual quality. In contrast, restoration-based methods like MAE better preserve fine textures, resulting in lower LPIPS.

## B.4 Model complexity comparison

A comparison of model complexity, including FLOPs, number of parameters, runtime, and memory usage, is summarized in Table 4.

## B.5 Stability at larger resolutions

In our benchmark, all image pairs are resized to 256×256 to ensure a fair comparison across methods. Additionally, we performed inference at a higher resolution of 512×512. The detailed quantitative results are shown in Table 5, and the accuracy trends indicate stable performance at larger input resolutions.

## C Physical constraints

To ensure a fair comparison, our main benchmark experiments adopted existing restoration and generation backbones with minimal modifications, avoiding additional priors that could introduce discrepancies across methods. Nevertheless, to address the concern of physical validity, we provide an additional analysis here by explicitly incorporating physically grounded constraints.

## C.1 DoP range constraint

The degree of polarization (DoP) is theoretically bounded within $[0, 1]$ [45, 6]. We enforce this property with a penalty loss:

$$\mathcal{L}_{\text{DoP-phys}} = \mathbb{E}\left[\max(0, \text{DoP} - 1)^2 + \max(0, -\text{DoP})^2\right], \tag{7}$$

which ensures physically valid DoP values and prevents over-polarization artifacts. This formulation illustrates how physically grounded constraints can be incorporated into learning frameworks to guide models toward valid polarization states, without introducing task-specific biases. In practice, adding this constraint yields slight but consistent improvements across both perceptual and polarization-related metrics.

## C.2 Potential extension: stokes consistency

Beyond DoP, polarization theory also imposes the inequality $S_1^2 + S_2^2 + S_3^2 \leq S_0^2$ [30], which guarantees consistency between the polarized and total intensities. While not included in our current benchmark implementation to maintain fairness, such constraints remain promising for future extensions.

## D Discussion

### D.1 Relationship between numerical metrics and stokes components

In our benchmark, PSNR and SSIM are used to evaluate pixel-level fidelity and structural similarity between the predicted and ground-truth Stokes components ($S_1$, $S_2$, $S_3$). LPIPS, on the other hand, provides a perceptual similarity score that captures differences in the spatial structure and appearance of the Stokes maps.

The Stokes components represent physical polarization information, and accurate reconstruction, reflected by high PSNR and SSIM, and low LPIPS, indicates better prediction quality. These metrics offer a quantitative assessment of how well the model captures the spatial and structural details of the polarization cues encoded in $S_1$, $S_2$, and $S_3$.

## E Limitations

Although our benchmark provides a comprehensive evaluation of RGB-to-polarization estimation, it still has several limitations due to practical constraints. First, our evaluation is based on the dataset provided by Jeon et al. [17]. While it includes more than 2000 scenes, the diversity of materials and surface types is limited. As a result, polarization estimation under complex material properties, such as metals, birefringent materials, and biological tissues, remains underexplored. Second, adverse conditions such as weather-induced degradations or extreme noise can degrade image quality and thus affect polarization estimation. Our benchmark does not evaluate these cases due to the lack of such scenarios in existing datasets. Expanding the dataset to include more challenging and diverse environments is an important direction for future work.

## F Societal impact

Polarization data has proven useful in many computer vision tasks, such as reflection separation and material classification. However, it remains difficult to obtain polarization data on consumer devices, limiting the practical use of polarization-based methods. This paper introduces a new task: RGB-to-polarization image estimation, which aims to infer polarization information directly from RGB images. By reducing reliance on specialized sensors, this approach has the potential to democratize access to polarization imaging and enable broader adoption in both scientific and industrial settings.

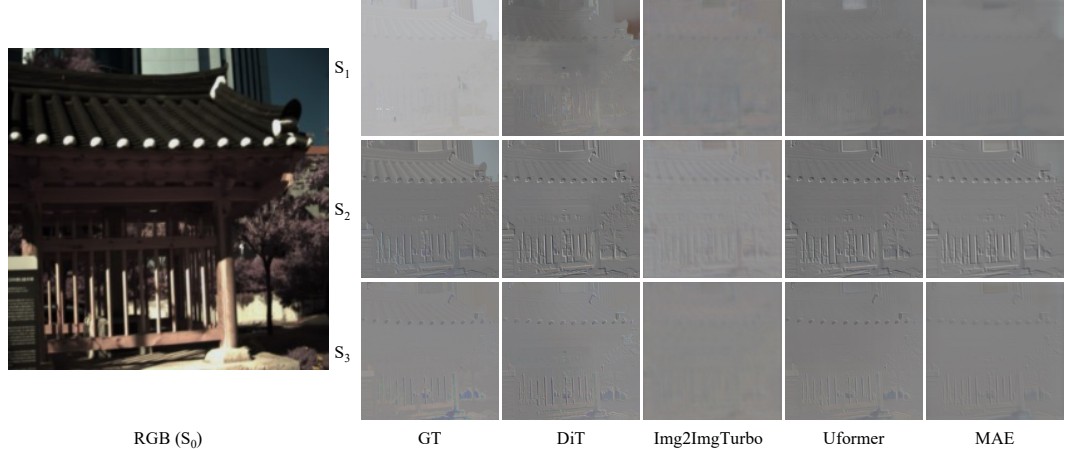

RGB (S$_0$)          GT          DiT          Img2ImgTurbo          Uformer          MAE

Figure 7: Qualitative comparison of estimated polarization components from RGB input. Results are shown for Uformer [47], MAE [14], DiT [36], and Img2ImgTurbo [35].

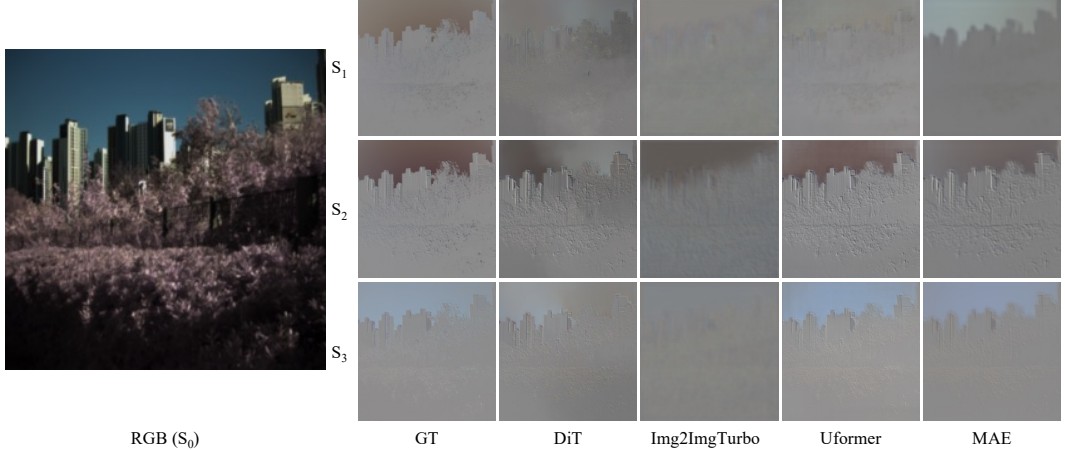

RGB (S$_0$)          GT          DiT          Img2ImgTurbo          Uformer          MAE

Figure 8: Qualitative comparison of estimated polarization components from RGB input. Results are shown for Uformer [47], MAE [14], DiT [36], and Img2ImgTurbo [35].

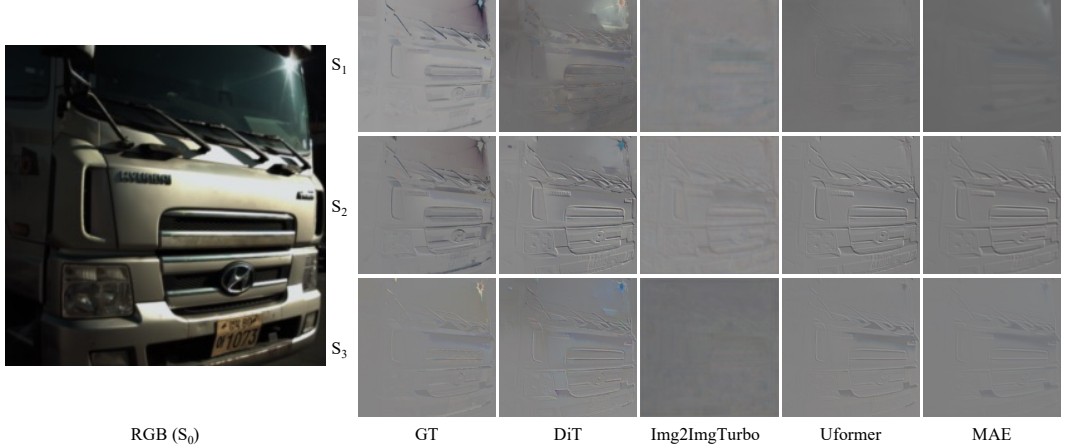

RGB (S$_0$)          GT          DiT          Img2ImgTurbo          Uformer          MAE

Figure 9: Qualitative comparison of estimated polarization components from RGB input. Results are shown for Uformer [47], MAE [14], DiT [36], and Img2ImgTurbo [35].

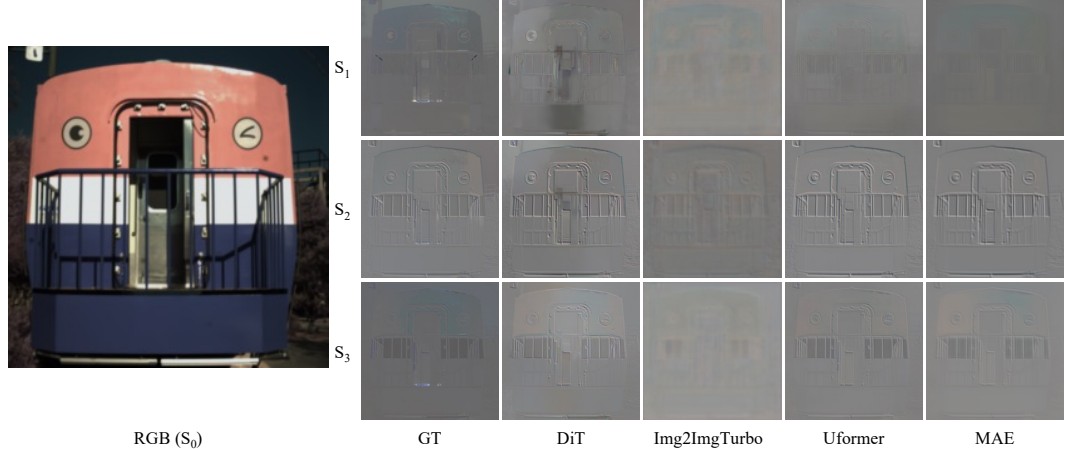

Figure 10: Qualitative comparison of estimated polarization components from RGB input. Results are shown for Uformer [47], MAE [14], DiT [36], and Img2ImgTurbo [35].

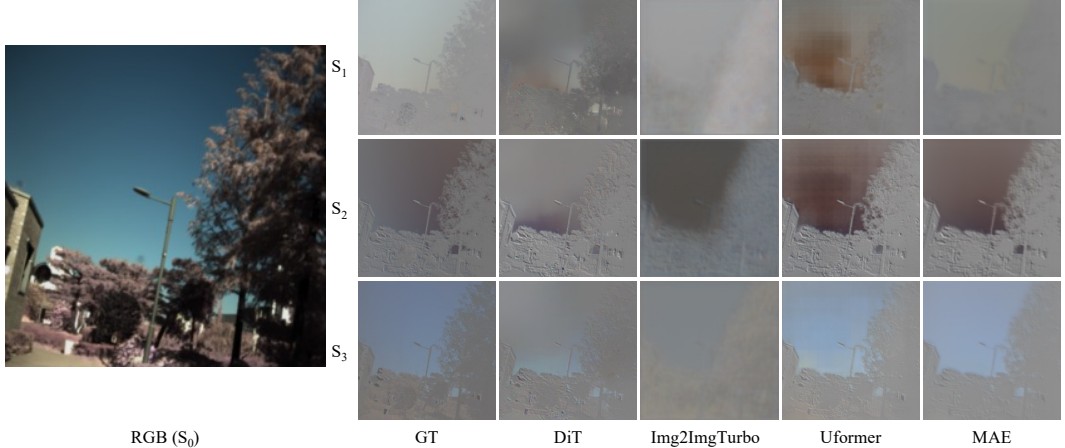

Figure 11: Qualitative comparison of estimated polarization components from RGB input. Results are shown for Uformer [47], MAE [14], DiT [36], and Img2ImgTurbo [35].

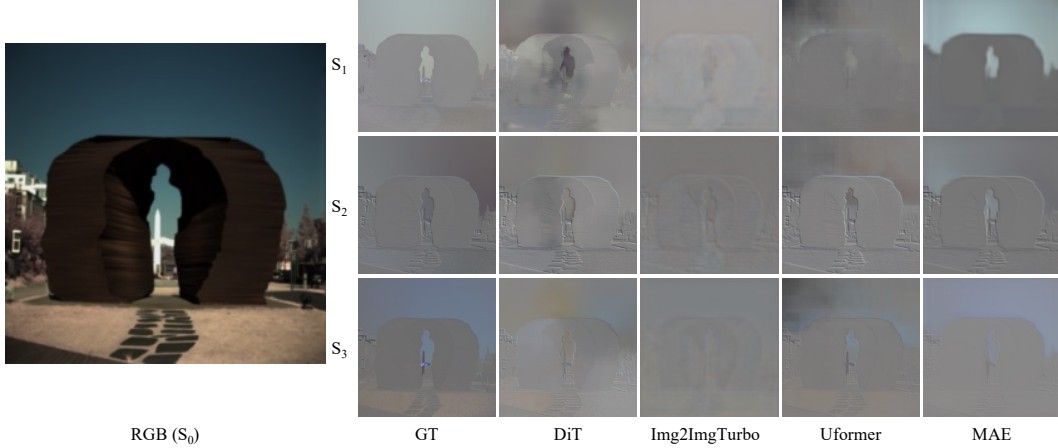

Figure 12: Qualitative comparison of estimated polarization components from RGB input. Results are shown for Uformer [47], MAE [14], DiT [36], and Img2ImgTurbo [35].

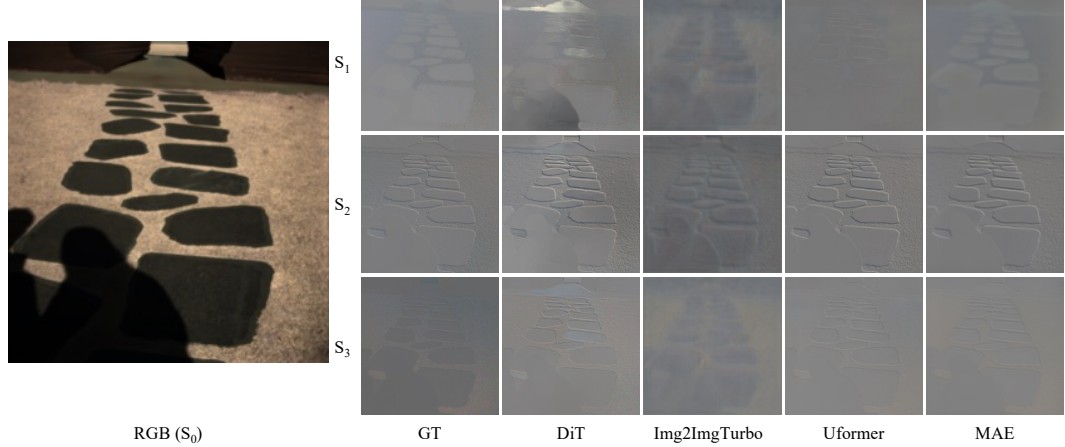

Figure 13: Qualitative comparison of estimated polarization components from RGB input. Results are shown for Uformer [47], MAE [14], DiT [36], and Img2ImgTurbo [35].

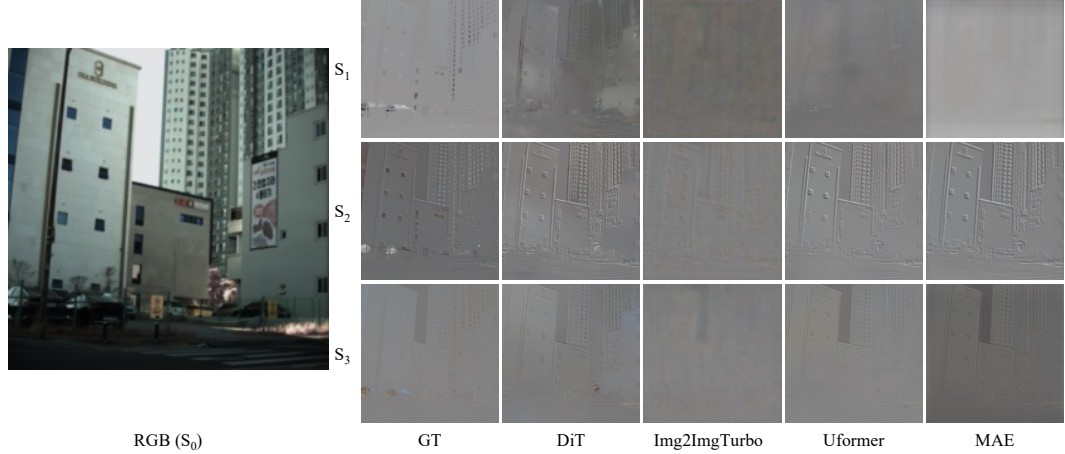

Figure 14: Qualitative comparison of estimated polarization components from RGB input. Results are shown for Uformer [47], MAE [14], DiT [36], and Img2ImgTurbo [35].

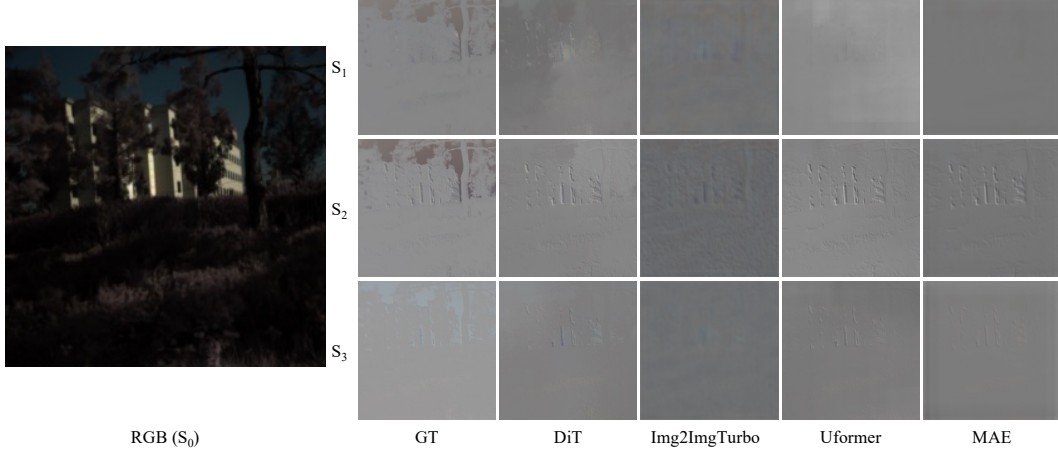

Figure 15: Qualitative comparison of estimated polarization components from RGB input. Results are shown for Uformer [47], MAE [14], DiT [36], and Img2ImgTurbo [35].

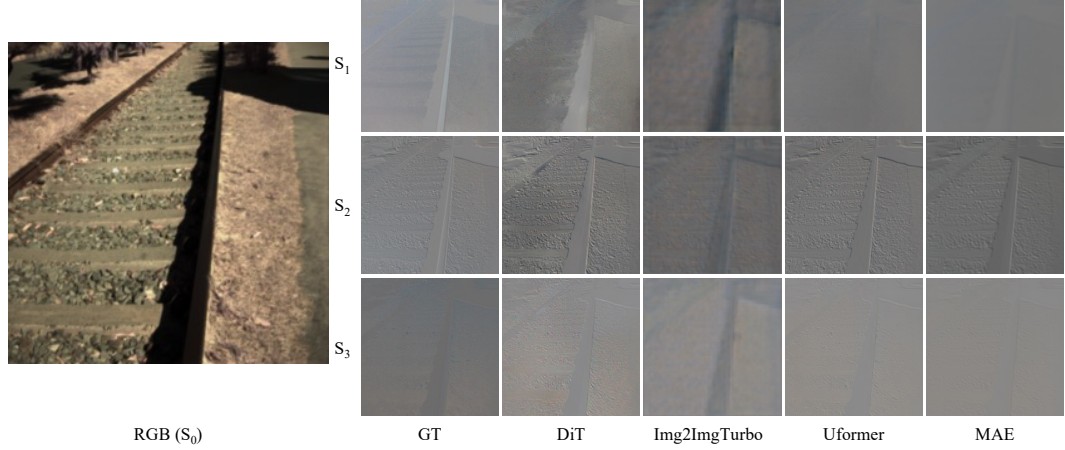

Figure 16: Qualitative comparison of estimated polarization components from RGB input. Results are shown for Uformer [47], MAE [14], DiT [36], and Img2ImgTurbo [35].

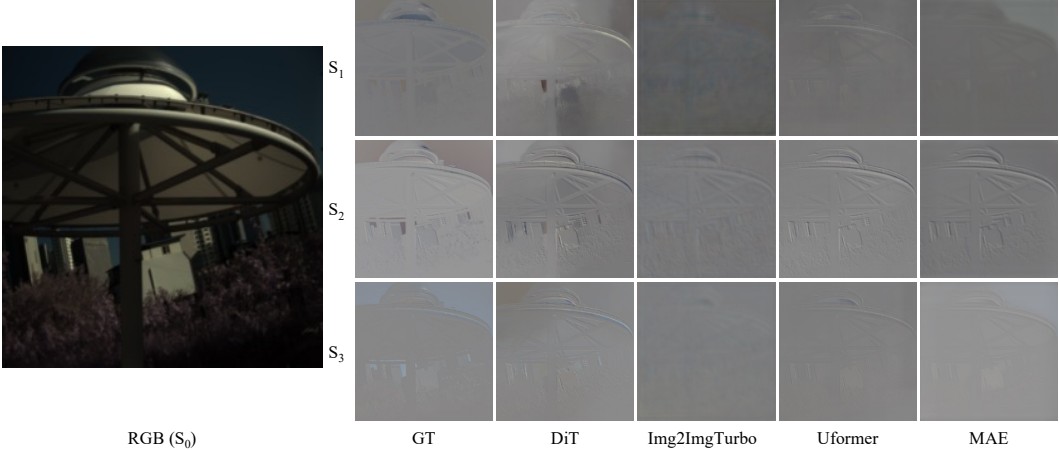

Figure 17: Qualitative comparison of estimated polarization components from RGB input. Results are shown for Uformer [47], MAE [14], DiT [36], and Img2ImgTurbo [35].

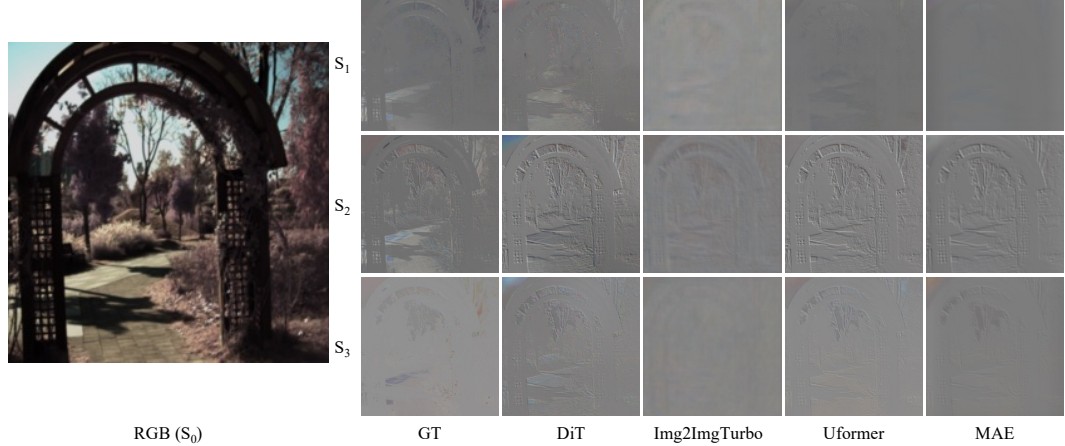

Figure 18: Qualitative comparison of estimated polarization components from RGB input. Results are shown for Uformer [47], MAE [14], DiT [36], and Img2ImgTurbo [35].

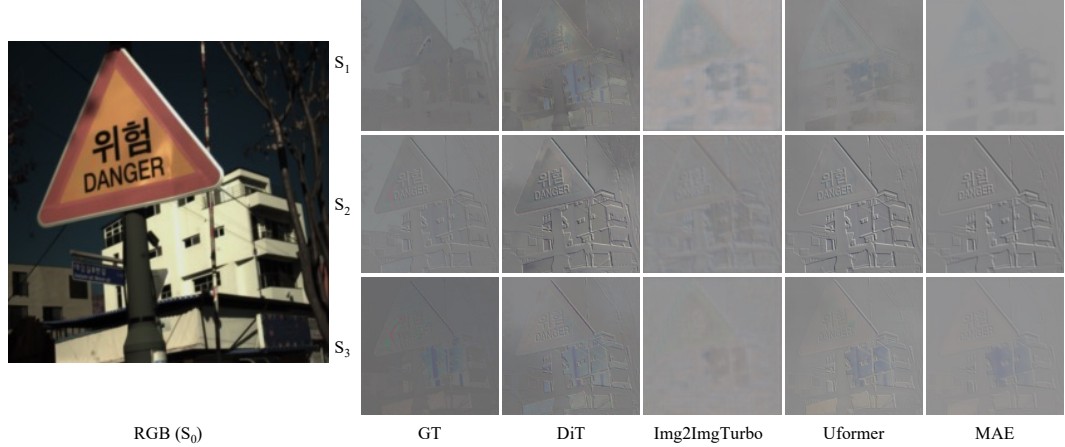

Figure 19: Qualitative comparison of estimated polarization components from RGB input. Results are shown for Uformer [47], MAE [14], DiT [36], and Img2ImgTurbo [35].

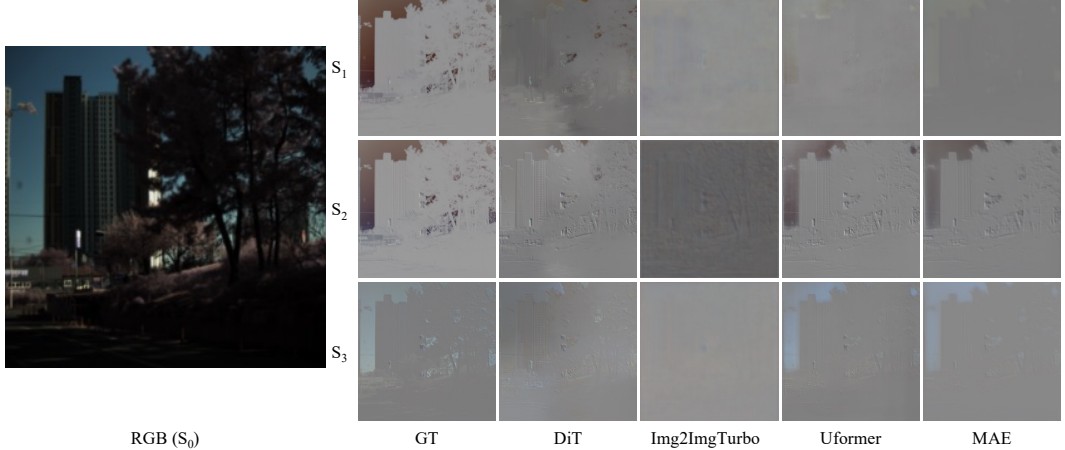

Figure 20: Qualitative comparison of estimated polarization components from RGB input. Results are shown for Uformer [47], MAE [14], DiT [36], and Img2ImgTurbo [35].

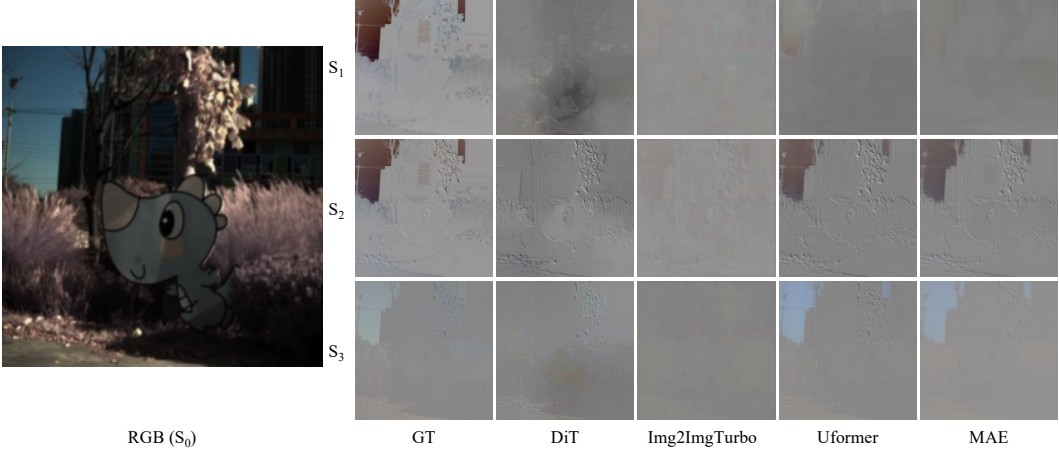

Figure 21: Qualitative comparison of estimated polarization components from RGB input. Results are shown for Uformer [47], MAE [14], DiT [36], and Img2ImgTurbo [35].

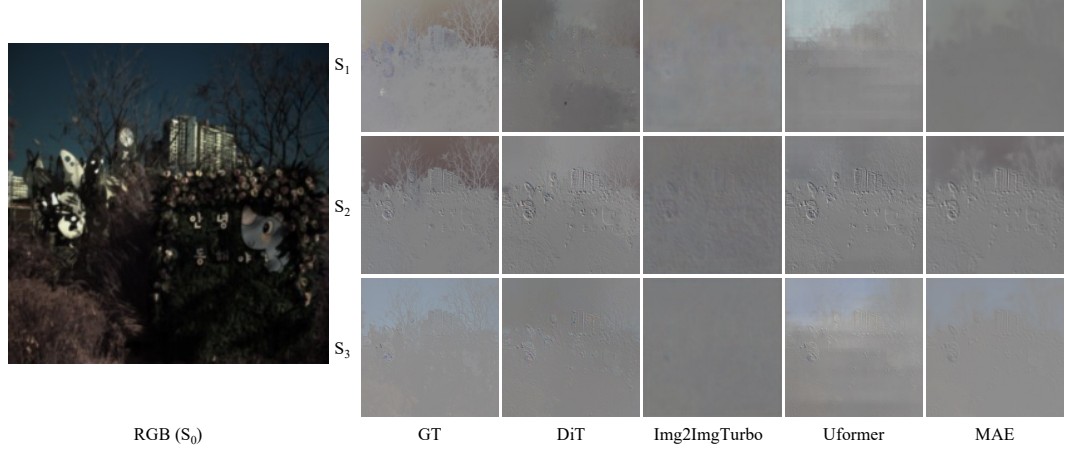

Figure 22: Qualitative comparison of estimated polarization components from RGB input. Results are shown for Uformer [47], MAE [14], DiT [36], and Img2ImgTurbo [35].

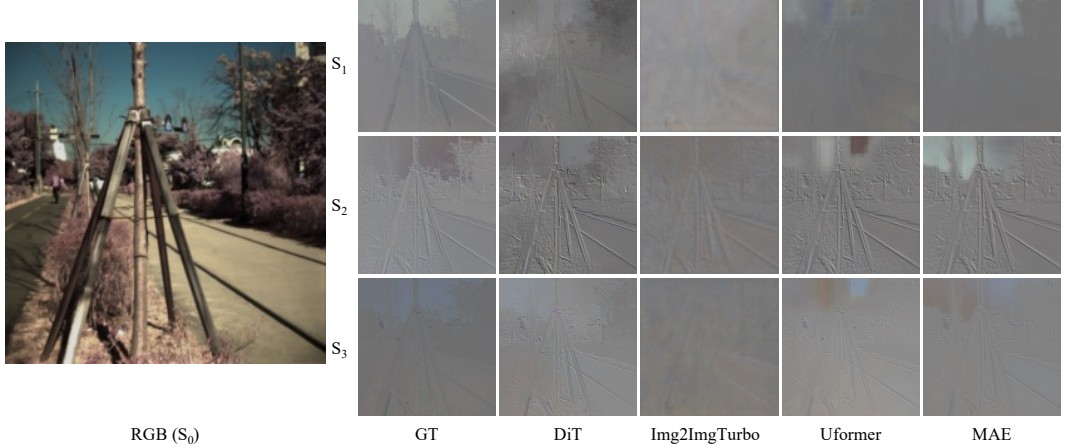

Figure 23: Qualitative comparison of estimated polarization components from RGB input. Results are shown for Uformer [47], MAE [14], DiT [36], and Img2ImgTurbo [35].

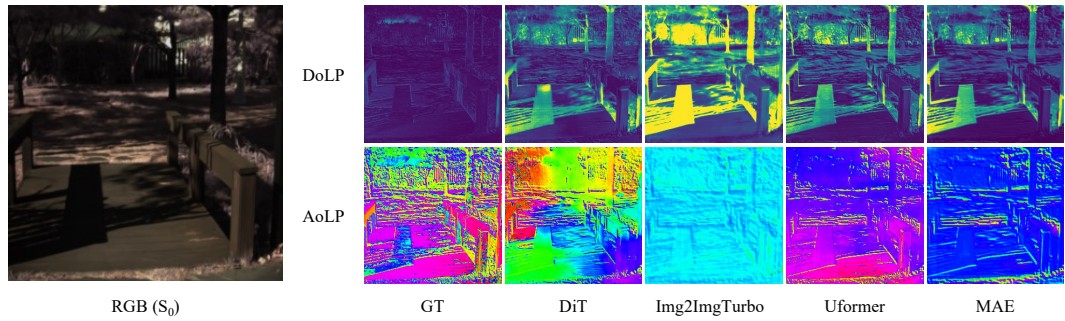

Figure 24: Qualitative comparison of predicted DoLP and AoLP maps, derived from Stokes components estimated from RGB input. Ground-truth maps are provided for reference, along with results from Uformer [47], MAE [14], DiT [36], and Img2ImgTurbo [35].

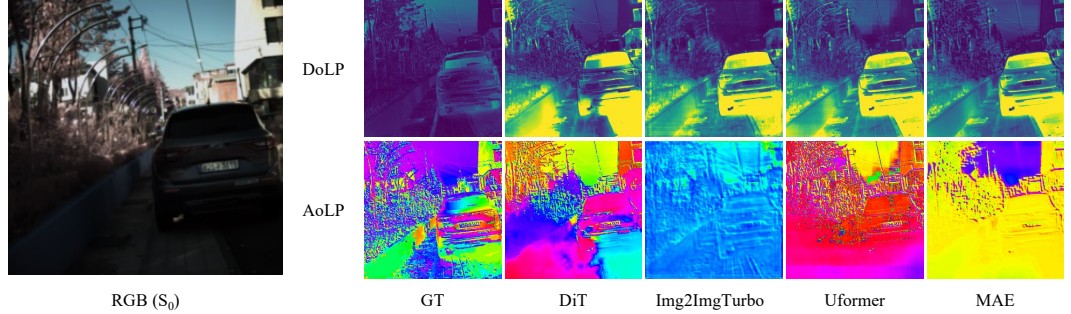

Figure 25: Qualitative comparison of predicted DoLP and AoLP maps, derived from Stokes components estimated from RGB input. Ground-truth maps are provided for reference, along with results from Uformer [47], MAE [14], DiT [36], and Img2ImgTurbo [35].

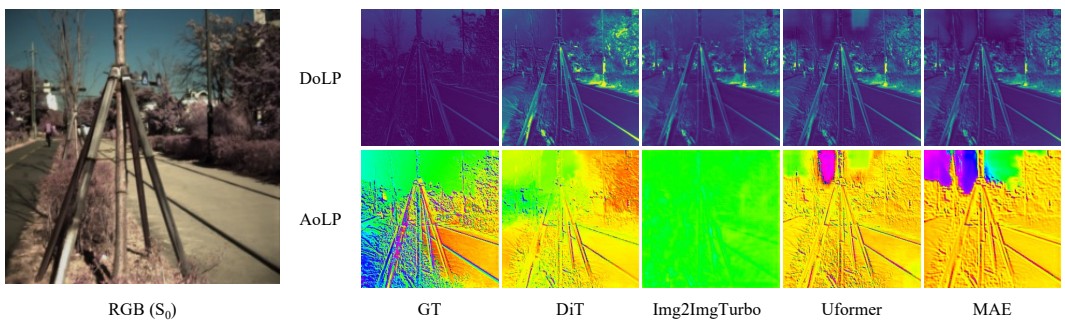

Figure 26: Qualitative comparison of predicted DoLP and AoLP maps, derived from Stokes components estimated from RGB input. Ground-truth maps are provided for reference, along with results from Uformer [47], MAE [14], DiT [36], and Img2ImgTurbo [35].

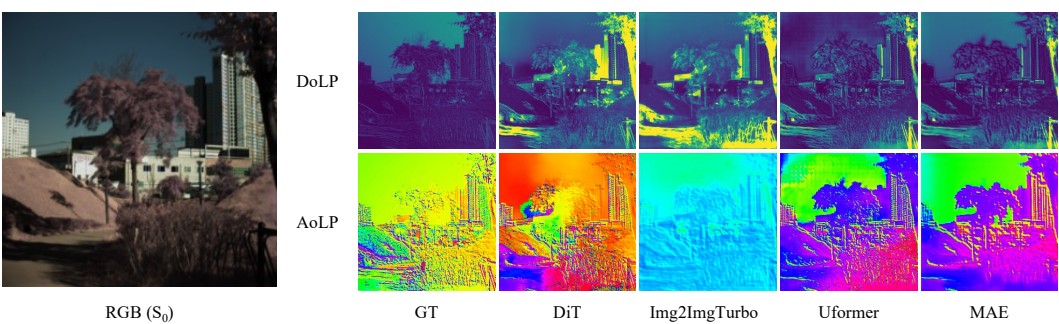

Figure 27: Qualitative comparison of predicted DoLP and AoLP maps, derived from Stokes components estimated from RGB input. Ground-truth maps are provided for reference, along with results from Uformer [47], MAE [14], DiT [36], and Img2ImgTurbo [35].

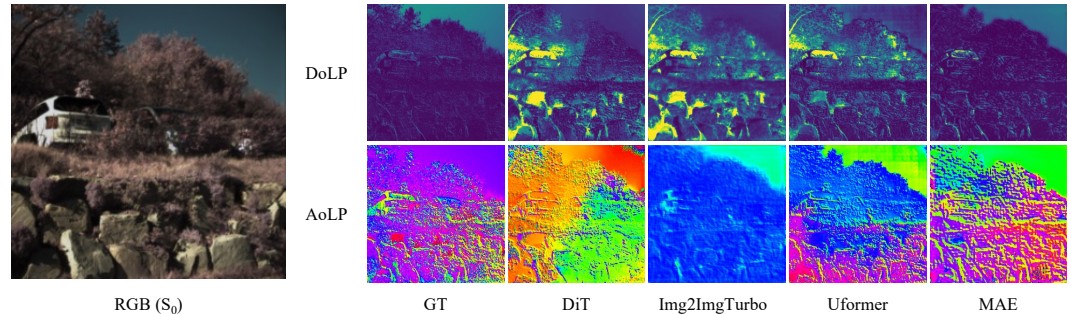

Figure 28: Qualitative comparison of predicted DoLP and AoLP maps, derived from Stokes components estimated from RGB input. Ground-truth maps are provided for reference, along with results from Uformer [47], MAE [14], DiT [36], and Img2ImgTurbo [35].

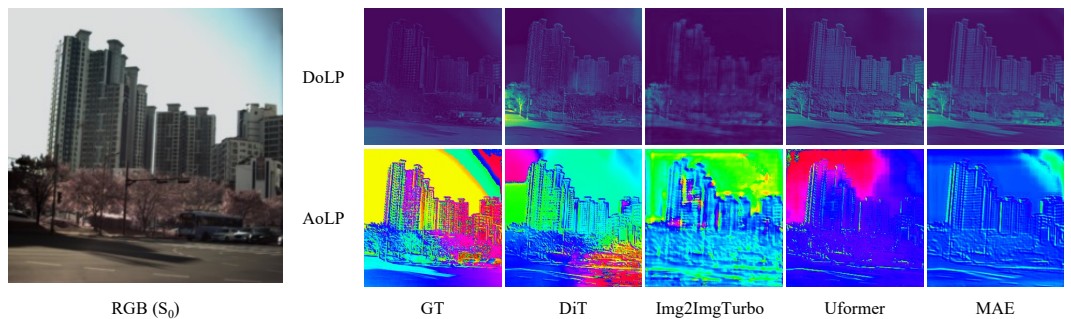

Figure 29: Qualitative comparison of predicted DoLP and AoLP maps, derived from Stokes components estimated from RGB input. Ground-truth maps are provided for reference, along with results from Uformer [47], MAE [14], DiT [36], and Img2ImgTurbo [35].

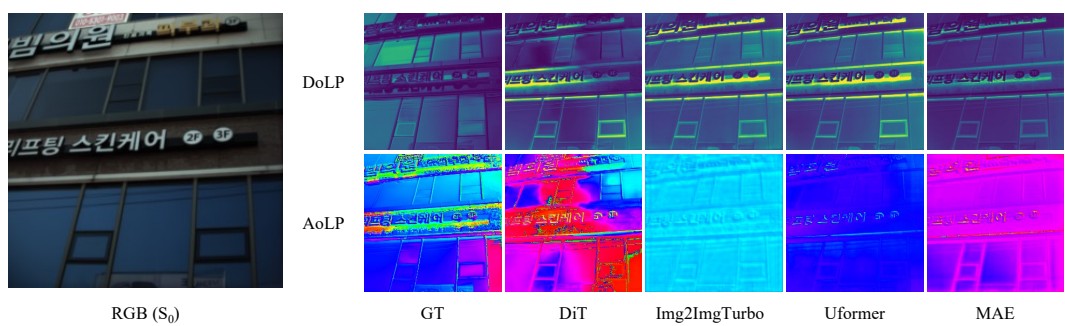

Figure 30: Qualitative comparison of predicted DoLP and AoLP maps, derived from Stokes components estimated from RGB input. Ground-truth maps are provided for reference, along with results from Uformer [47], MAE [14], DiT [36], and Img2ImgTurbo [35].

