# OpenReview forum: "RGB-to-Polarization Estimation: A New Task and Benchmark Study"
_NeurIPS.cc/2025/Datasets_and_Benchmarks_Track — NeurIPS 2025 Datasets and Benchmarks Track poster_

### Official Review · Reviewer_Mrsg · 2025-06-10

**Rating:** 4
**Confidence:** 5

**Summary:**

This manuscript introduces a novel task of estimating polarization images from RGB images without the need for optical components. It establishes the first benchmark with evaluation protocols and provides comparisons with related methods. While the task and benchmarks are valuable and the proposed methods are innovative, the experiments are conducted on a single public dataset, which limits the diversity and robustness of the evaluation.

**Additional Feedback:**

1. Adding downstream task experiments is highly recommended to demonstrate the practical utility and broader impact of the proposed method.
2. Consider incorporating and addressing physical constraints to enhance the model's validity and ensure consistency with real-world physics.
3. Strengthen the qualitative analysis by explicitly linking the results to physical phenomena for improved interpretability and relevance.
4. Some state-of-the-art monocular depth estimation methods could be assessed and benchmarked for polarization prediction.
5. It would be nice to discuss the stability of the prediction at a larger resolution of polarization images.
6. More polarization-specific evaluation metrics could be discussed and explored for assessing the reliability of the estimation.

**Dataset Code Accessibility:**

Yes

**Dataset Code Comments:**

The dataset and code are accessible, well-documented, and provided in their final, executable form. The authors include sufficient details to ensure reproducibility, making the resources easy to use for the research community.

**Ethical Comments:**

No significant ethical concerns remain because the study uses publicly available datasets with no evidence of privacy, copyright, or consent violations. Additionally, the research focuses on technical advancements and poses no societal, safety, or environmental risks.

**Ethical Considerations:**

No, there are no or only very minor ethics concerns

**Final Justification:**

The response helps to clarify many concerns. More methods are benchmarked, and the differences between the present work and existing relevant works are discussed in detail. For these reasons, we would like to raise the rating.

**Limitations Weaknesses:**

1. The benchmark relies on a single dataset, raising concerns about the generalizability of the results to other polarization data. The authors should consider collecting additional RGB-polarization pair data using real sensors to enhance the robustness of the evaluation.

2. The paper does not demonstrate how the estimated polarization images can enhance practical tasks and analyze the performance gap compared to real polarization images.

3. The baseline lacks physical constraints for the degree of polarization (DOP). Furthermore, the evaluation metrics should include physical consistency and perceptual quality to enhance realism.

4. The paper should include a brief review of existing RGB-to-polarization methods. Summarizing prior work, such as [#1], [#2], and [#3], would help highlight the advancements introduced in this study. It is unfair to claim the proposed benchmark as the first benchmark, as polarization estimation has already been researched. The problematic claim should be revised.

   [#1] Sharecmp: Polarization-aware RGB-P Semantic Segmentation, 2025

   [#2] Simultaneous Acquisition of High-Quality RGB Image and Polarization Information Using a Sparse Polarization Sensor, 2023

   [#3] Predicting Polarization Beyond Semantics for Wearable Robotics, 2018

5. It would be nice to discuss the prediction of different polarization information, like DoLP and AoLP.

6. It would be nice to also report the efficiency results of polarization estimation, like FLOPs/MACs, the number of parameters, running time, and memory requirements.

**Strengths Contributions:**

1. The paper studies a meaningful topic by framing polarization imaging as an estimation task using standard RGB images.
2. It presents the first benchmark with public datasets, clear evaluation metrics, and strong baseline models to enable fair comparisons.
3. The study conducts a comprehensive evaluation of various model types, including restoration-based, generative, and pre-trained models, offering valuable insights into their performance.
4. The analysis is thoughtful, and the authors propose future directions, including the incorporation of physical constraints and the exploration of self-supervised learning.
5. The manuscript is well-written and nicely organized, and provides clear explanations of the methods and experiments.

---

> ### Author Rebuttal · Authors · 2025-07-30
>
> **W1 Generalizability**
>
> Following the suggestion, we conducted additional experiments on other RGB-polarization datasets, including that of Qiu et al. (2021) and the RSP Dataset [Kurita et al., 2024]. The corresponding results are provided below and will be incorporated into the revised manuscript.
>
> | **Method** | **Qiu PSNR** | **Qiu SSIM** | **Qiu LPIPS** | **RCP PSNR** | **RCP SSIM** | **RCP LPIPS** |
> | ---------- | ------------ | ------------ | ------------- | ------------ | ------------ | ------------- |
> | Restormer  | 14.77        | 0.5325       | 0.5186        | 18.8500      | 0.8394       | 0.2697        |
> | Uformer    | 14.68        | 0.7278       | 0.3160        | 18.7350      | 0.8381       | 0.2627        |
> | Wdiff      | 11.44        | 0.6523       | 0.4042        | 11.5600      | 0.6945       | 0.4311        |
> | DiT        | 14.74        | 0.7328       | 0.2829        | 17.8550      | 0.8191       | 0.2874        |
> | Realfill   | 15.19        | 0.7241       | 0.3028        | 18.09 	 	            | 0.8252            | 0.2654        |
> | ImgTurbo   | 15.65        | 0.7566       | 0.3811        | 18.7800      | 0.8504       | 0.3115        |
> | MAE    | 15.02        | 0.7401       | 0.3238        | 18.8050      | 0.8499       | 0.2772        |
>
> [#1] Linear Polarization Demosaicking for Monochrome and Colour Polarization Focal Plane Arrays, Computer Graphics Forum*, 2021, Qiu et al.
>
> [#2] Simultaneous Acquisition of High Quality RGB Image and Polarization Information using a Sparse Polarization Sensor*, WACV 2024, Kurita et al.
>
>
> **W2/Q1 Downstreaming Tasks**
>
> Following the suggestion, we conducted preliminary experiments on the RGB-P semantic segmentation task [#3], following its settings and using the ShareCMP framework with the UPLight dataset. As summarized below, segmentation performance degrades when using estimated polarization inputs, highlighting both the current limitations and the potential of this direction for future development. We will include the results in the revised version.
>
> | I₀       | I₄₅       | I₉₀       | I₁₃₅      | MIoU ↑  |
> |----------|-----------|-----------|-----------|--------|
> | real     | real      | real      | real      | 92.45  |
> | estimated | real     | real      | real      | 76.91  |
> | real     | estimated | real      | real      | 92.40  |
> | real     | real      | estimated | real      | 57.40  |
> | real     | real      | real      | estimated | 75.68  |
> | estimated | estimated | estimated | estimated | 45.56 |
>
> [#3] Sharecmp: Polarization-aware RGB-P Semantic Segmentation, 2025
>
> **W3/Q2 Physical Constraints**
>
> To ensure a fair comparison, our benchmark initially adopted existing restoration and generation backbones with minimal modifications, avoiding added physical constraints that could introduce discrepancies and hinder a unified evaluation. Moreover, generation-based frameworks such as diffusion models estimate noise maps during training, which makes it challenging to incorporate physical priors in a straightforward manner.
>
> However, to address this concern, we have explicitly incorporated physically grounded constraints into our benchmark pipeline. Specifically, based on a state-of-the-art restoration model (MAE), we introduce a DoP range loss to demonstrate the effect of incorporating physical constraints [#4, #5]:
>
> - **DoP Range Constraint**: a penalty loss that encourages the predicted DoP to stay within the physically valid range $[0, 1]$:
>
> $$
> \mathcal{L}_{\text{DoP-phys}} = \mathbb{E} \left[ \max(0, \text{DoP} - 1)^2 + \max(0, -\text{DoP})^2 \right]
> $$
>
> The physical constraints along with the relevant physical discussion will be detailed in the newly added **Section 3.2 "Physical Constraints"** (line 151 of the original manuscript).
>
> The experimental results are summarized below. We find that incorporating physical constraints into the MAE framework yields consistent improvements across all perceptual and physically relevant metrics. Specifically, the LPIPS score decreases from 0.2684 to 0.2672, indicating slightly better perceptual similarity. More notably, the DoP error is reduced from 2.7625 to 2.7244.
>
> | Method     | PSNR ↑  | SSIM ↑  | LPIPS ↓  | DoP ↓  |
> |------------|--------|--------|--------|--------|
> | MAE    | 24.74  | 0.8876 | 0.2684 | 2.7625 |
> | MAE+physical constraints    | 24.65  | 0.8869 | 0.2672 | 2.7244 |
>
> [#4: J. Scott Tyo et al., *Applied Optics*, 2006]
>
> [#5: Russell A. Chipman, *Applied Optics*, 2005]
>
> **W3/Q6 Polarization-specific Evaluation and Perceptual Quality**
>
> In our benchmark, we use the commonly adopted perceptual quality metric LPIPS, as shown in Table 1.
> Following the suggestions, we additionally include a polarization-specific evaluation by estimating the DoP. The experimental results are presented below (the lower the better).
>
>
> | Metrics     | Restormer | Uformer | Wdiff  | DiT    | Realfill | ImgTurbo | **MAE** |
> |------------|-----------|---------|--------|--------|-----------|-----------|---------|
> | DoP       | 2.750    | 2.764  | 3.069 | 2.738 | 2.817    | 2.794    | 2.763  |
>
>
> **W4 Review of existing polarization estimation methods**
>
> In the revised manuscript, we will include a discussion of representative efforts in Section 2 “Related Work” (line 58), including ShareCMP (Liu et al., 2025), Kurita et al. (2023), and Yang et al. (2018). We will revise the claim to properly acknowledge prior work and clarify the differences.
>
> **W5 Discussion on DoLP and AoLP**
>
> We will add a new subsection in Section 3.1 to define key derived polarization quantities (e.g., DoLP, AoLP) and explain their physical meanings. Moreover, to improve interpretability, we will also include corresponding visualization channels in the revised manuscript. Due to this year's rebuttal policy, we are unable to include additional figures at this stage. However, we are committed to incorporating these visualizations in the final camera-ready version.
>
> **W6 Efficiency**
>
> The FLOPs (or MACs), number of parameters, runtime, and memory usage of different methods are summarized below.  The runtime is measured on a single NVIDIA RTX A5000 GPU.
> | Method     | FLOPs     | Trainable Parameters | Running Time | Training Memory   | Test Memory |
> |------------|-----------|----------------------|---------------|-------------------|-------------|
> | Restormer  | 38 GFLOPs | 26.1M                | 0.04s         | 2 × 24G GPUs      | ~1G         |
> | Uformer    | 11 GFLOPs | 5.3M                 | 0.01s         | 2 × 24G GPUs      | ~1G         |
> | Wdiff      | 16 GFLOPs | 109.7M               | 0.25s         | 4 × 24G GPUs      | ~1G         |
> | DiT        | 19 GFLOPs | 108.7M               | 0.19s         | 4 × 24G GPUs      | ~1G         |
> | Realfill   | 86 GFLOPs | 1.7M                | 5.00s         | 4 × 24G GPUs      | ~3G         |
> | ImgTurbo   | 86 GFLOPs | 9.5M                | 0.20s         | 4 × 24G GPUs      | ~6G         |
> | MAE    | 64 GFLOPs | 330.0M                 | 0.01s         | 4 × 24G GPUs      | ~2G         |
>
> These metrics will be included in the revised version.
>
> **Q3 Physical Phenomena**
>
> In our RGB-to-polarization estimation task, we find that estimating the $S_1$ component is more difficult than $S_2$ and $S_3$. Physically, $S_1$ represents the difference between horizontal and vertical polarization, which tends to be more sensitive to surface orientation and material properties. In many natural scenes, this component is weaker or more spatially uniform due to diffuse reflection, leading to a lower signal-to-noise ratio and making it harder for the model to learn accurate patterns from RGB inputs. This will be added in the revised manuscript.
>
> **Q4 Depth Estimation**
>
> We have conducted experiments using Depth Anything v2 [#6]. Specifically, we modified the last layer to output 9 channels for predicting $S_1$, $S_2$ and $S_3$, while keeping the pre-trained parameters for the remaining layers. The averaged PSNR, SSIM, and LPIPS are 25.75, 0.8777, and 0.3765, respectively. Although this model achieves higher pixel-level metrics compared to MAE, the perceptual quality is noticeably worse. This suggests that pre-trained depth-based models provide strong structural priors and achieve better pixel-level metrics. However, their outputs tend to be over-smoothed, leading to worse perceptual quality. In contrast, restoration-based methods like MAE better preserve fine textures, resulting in lower LPIPS.
> [#6] Yang, Lihe, et al. "Depth Anything v2." NeurIPS, 2024.
>
> **Q5 Stability at larger resolutions**
>
> In our benchmark, all image pairs are resized to 256×256 to ensure a fair comparison across methods. Additionally, we performed inference at a higher resolution of 512×512. The detailed quantitative results are shown below, and the accuracy trends indicate stable performance at larger input resolutions. We will include this discussion in the revised manuscript.
>
> | Method    | PSNR@256  | SSIM@256| LPIPS@256 | PSNR@512 | SSIM@512 | LPIPS@512 |
> |-----------|----------------|----------------|------------------|----------------|----------------|------------------|
> | Restormer | 23.98          | 0.8730         | 0.2369           | 23.35          | 0.8757         | 0.2379           |
> | Uformer   | 24.34          | 0.8722         | 0.2267           | 24.11          | 0.8883         | 0.2092           |
> | Wdiff     | 13.11          | 0.6822         | 0.3772           | 12.53          | 0.6765         | 0.3997           |
> | DiT       | 23.33          | 0.8409         | 0.2447           | 18.65          | 0.8303         | 0.2394           |
> | Realfill  | 21.81          | 0.8025         | 0.2654           | 23.74          | 0.8238         | 0.2349           |
> | ImgTurbo  | 23.33          | 0.8725         | 0.3542           | 22.52          | 0.8372         | 0.3602           |
> | MAE       | 24.74          | 0.8876         | 0.2684           | 23.63          | 0.8650         | 0.2810           |

---

> > ### Comment · Reviewer_Mrsg · 2025-08-03
> > **Comment**
> >
> > The rebuttal helps to clarify many concerns.
> >
> > However, the revised claim should be directly added in the rebuttal to acknowledge previous research works on polarization estimation, and the differences should be clarified and discussed in detail.

---

> > > ### Author Response · Authors · 2025-08-03
> > >
> > > Thank you for the valuable feedback. Due to word limitations in the original rebuttal, we were unable to elaborate on the revised claim. We now provide a more detailed clarification in response to this point.
> > >
> > > We acknowledge that prior works have explored polarization estimation in various forms. In the revised manuscript, we will include a discussion of representative efforts in Section 2 “Related Work” (line 58 of the original manuscript), including ShareCMP (Liu et al., 2025), Kurita et al. (2023), and Yang et al. (2018). While these studies have made important contributions to related areas such as task-specific polarization estimation, they differ from a standardized benchmark targeting RGB-to-Stokes polarization estimation. To clarify, we will revise our wording in the manuscript to appropriately credit prior work, and we will summarize their main contributions and highlight key differences from our approach as outlined below.
> > >
> > > ---
> > > 1. ShareCMP (Liu et al., 2025) introduces an RGB–polarization semantic segmentation framework based on a shared dual-branch architecture. It takes four-direction polarized images ($I_{0}$, $I_{45}$, $I_{90}$, $I_{135}$) as input and generates polarization-modal images ($I_p$) via a Polarization Generation Attention (PGA) module. The generated $I_p$ is optimized end-to-end for the segmentation task, making it more conducive to accurate segmentation.
> > >
> > >     **Differences:** In contrast, our work focuses on establishing a standardized benchmark, where models are evaluated based on their ability to predict full Stokes components ($S_1$, $S_2$, $S_3$) directly from RGB inputs. This formulation enables a model-agnostic evaluation of polarization reconstruction performance.
> > >
> > > ---
> > > 2. Kurita et al. (2023) take the raw image obtained from a sparse polarization sensor as input and aim to generate high‑quality RGB images and polarization information.
> > >
> > >
> > >     **Differences:** Unlike Kurita et al. (2023), we propose a standardized evaluation benchmark that assesses model performance in estimating the full Stokes components directly from RGB inputs.
> > >
> > > ---
> > > 3. Yang et al. (2018) propose predicting polarization information ($I_{0}$ and $I_{90}$) from RGB images for semantic segmentation in wearable robotics. (As clarified by the authors in Section IV, Experiments: “We retrofit it by attaching horizontal and vertical polarization filters on the left and right camera, respectively.”)
> > >
> > >     **Differences:** In contrast, our method focuses on benchmarking RGB-to-polarization estimation. Unlike Yang et al. (2018) which predicts only $S_1$ derived from $I_{0}$ and $I_{90}$, our benchmark aims to estimate the full set of Stokes components ($S_1$, $S_2$, $S_3$), enabling a more comprehensive evaluation. Furthermore, our benchmark assesses a diverse range of frameworks, including both restoration-based and generation-based backbones. This analysis highlights the strengths and limitations of existing methods and suggests potential directions for future research in the field.
> > >
> > > ---
> > > We thank the reviewer again for the constructive suggestion. In the revised manuscript, we will include the above comparative discussions to properly acknowledge related efforts and clarify how our benchmark complements them. We hope this addition helps position our work more clearly within the existing literature.

---

> > > > ### Comment · Reviewer_Mrsg · 2025-08-04
> > > > **Comment**
> > > >
> > > > The reviewer would like to thank the authors for the response with the revised claim, which should be added in the final version to acknowledge these relevant previous research works.

---

> > > > > ### Author Response · Authors · 2025-08-07
> > > > >
> > > > > We sincerely thank the reviewer for their encouraging follow-up. As suggested, we will incorporate the revised claim and the detailed comparative discussion into the final version to appropriately acknowledge prior works.

---

### Official Review · Reviewer_awtX · 2025-06-20

**Rating:** 4
**Confidence:** 4

**Summary:**

This paper proposes a benchmark study for generating polarization images from conventional RGB images without requiring specialized sensors. By leveraging existing RGB-polarization datasets, this paper conducts extensive experiments using various deep-learning-based approaches, particularly restoration-based and generative methods. Based on the benchmark results, the paper provides insights into future model design and identifies key open challenges in estimating polarization information from RGB images.

**Additional Feedback:**

Most of the feedback have been reflected in both the strengths and weaknesses.

**Dataset Code Accessibility:**

Yes

**Dataset Code Comments:**

Benchmark code is available at https://anonymous.4open.science/r/Polarization2RGB-66F0/

**Ethical Comments:**

This study builds upon publicly available datasets and code.

**Ethical Considerations:**

No, there are no or only very minor ethics concerns

**Final Justification:**

The technical contribution and novelty of this paper were already sufficient, and the authors have provided sufficient rebuttals regarding the visualization, related work, and dataset. Therefore, I remain my original score.

**Limitations Weaknesses:**

1. The experiments are limited to a single dataset [Jeon et al., 2024], which was captured using a polarization camera. However, images captured by polarization cameras generally exhibit lower quality compared to datasets constructed using standard RGB cameras combined with polarization rotations. Given that the paper focuses on estimating polarization information from RGB inputs, it would be more meaningful to evaluate performance on datasets captured with conventional RGB cameras. The absence of experiments across more diverse domains, such as those provided by [Qiu et al., 2021] and the RSP Dataset [Kurita et al., 2024; missing reference] represents a main limitation.

2. The qualitative results of the reconstructed polarization images are presented only through the visualization of the $s_1$, $s_2$, and $s_3$ Stokes components. While these components represents the polarization properties, they are not well-suited for evaluating the fine detailed visual reconstruction quality. The visualization strategy could be significantly improved by adopting the representations used in other polarization methods [Jeon et al., 2024], such as Angle of Linear Polarization (AoLP), Degree of Polarization (DoP), and Circular Degree of Polarization (CoP), along with appropriate colormaps to enable more detailed and interpretable analysis.

3. The following references are missing and should be included in the manuscript.
* Real and synthetic polarization datasets
     * Simultaneous Acquisition of High Quality RGB Image and Polarization Information using a Sparse Polarization Sensor*, WACV 2024, Kurita et al.
* Polarization-based applications that leverage AoLP and DoP for more accurate geometry and appearance estimation than RGB alone:
     * Simultaneous Acquisition of Polarimetric SVBRDF and Normals, SIGGRAPH Asia 2018, Baek et al.
     * Sparse Ellipsometry: Portable Acquisition of Polarimetric SVBRDF and Shape with Unstructured Flash Photography, SIGGRAPH 2022, Hwang et al.
     * Polarimetric BSSRDF Acquisition of Dynamic Faces, SIGGRAPH Asia 2024, Ha et al.

Although these issues are critical, I am leaning toward positive based on the overall contribution of the paper. However, the major weakness should be clearly addressed during the rebuttal.

**Strengths Contributions:**

The paper provides a comprehensive benchmark of various deep learning-based methods for estimating polarization images from RGB inputs, motivated by the potential advantages of polarization imaging. Notably, paper includes a various range of model architectures, such as state-of-the-art diffusion-based approaches leveraging pretrained networks on large-scale datasets, as well as methods based on recent Vision Transformer architectures. This benchmark is expected to serve as a valuable reference for future research aimed at reconstructing polarization information from RGB images.

---

> ### Author Rebuttal · Authors · 2025-07-30
>
> **W1 Extension to diverse datasets**
>
> Many thanks for the suggestion. Following the suggestion, we have extended our experiments to more diverse datasets, including the one from Qiu et al. (2021) and the RSP Dataset [Kurita et al., 2024], to better assess the generalization capability of our method. The corresponding results are provided below and will be incorporated into the revised manuscript.
>
> | **Method** | **Qiu PSNR** | **Qiu SSIM** | **Qiu LPIPS** | **RCP PSNR** | **RCP SSIM** | **RCP LPIPS** |
> | ---------- | ------------ | ------------ | ------------- | ------------ | ------------ | ------------- |
> | Restormer  | 14.77        | 0.5325       | 0.5186        | 18.8500      | 0.8394       | 0.2697        |
> | Uformer    | 14.68        | 0.7278       | 0.3160        | 18.7350      | 0.8381       | 0.2627        |
> | Wdiff      | 11.44        | 0.6523       | 0.4042        | 11.5600      | 0.6945       | 0.4311        |
> | DiT        | 14.74        | 0.7328       | 0.2829        | 17.8550      | 0.8191       | 0.2874        |
> | Realfill   | 15.19        | 0.7241       | 0.3028        | 18.09 	 	            | 0.8252            | 0.2654        |
> | ImgTurbo   | 15.65        | 0.7566       | 0.3811        | 18.7800      | 0.8504       | 0.3115        |
> | MAE    | 15.02        | 0.7401       | 0.3238        | 18.8050      | 0.8499       | 0.2772        |
>
>
> **W2 Suggestions on Visualizations**
>
> Thank you for the insightful suggestion. Following your recommendation, we will include additional visualization channels in the revised version of our manuscript. Specifically, we will add the Angle of Linear Polarization (AoLP), Degree of Polarization (DoP), and Circular Degree of Polarization (CoP), each rendered with appropriate colormaps to enhance interpretability and facilitate more detailed visual comparisons.
>
> Due to this year's rebuttal policy, we are unable to include additional figures at this stage. However, we are committed to incorporating these visualizations in the final camera-ready version.
>
> In addition, to provide a clearer theoretical grounding for these derived representations, we have updated Section 3.1 of the manuscript (line 150) with a new subsection titled *Interpretable Polarization Features*. This addition offers mathematical definitions of DoP, DoLP, CoP, and AoLP, and explains their perceptual advantages over raw Stokes components. The new content is illustrated below and will be integrated into the revised manuscript:
>
> $$
> \text{DoP} = \frac{\sqrt{S_1^2 + S_2^2 + S_3^2}}{S_0} = \sqrt{\text{DoLP}^2 + \text{CoP}^2}
> $$
>
> $$
> \text{DoLP} = \frac{\sqrt{S_1^2 + S_2^2}}{S_0}, \quad
> \text{CoP} = \frac{S_3}{S_0}, \quad
> \text{AoLP} = \frac{1}{2} \arctan\left(\frac{S_2}{S_1}\right)
> $$
>
> These equations serve as interpretable alternatives to the raw Stokes components, with their physical implications to be further detailed in the revised version of the manuscript.
>
>
>
> **W3 References**
>
> Following the suggestions, we will add the relevant references and accompanying discussions in the revised version. Section 2.1 (line 74) on polarization datasets will be expanded to include additional representative datasets, including the one introduced by Kurita et al. (WACV 2024). In Section 2.2 (line 113), under Polarization-related tasks, we will also incorporate more works such as Baek et al. (SIGGRAPH Asia 2018), Hwang et al. (SIGGRAPH 2022), and Ha et al. (SIGGRAPH Asia 2024), along with corresponding descriptions to better reflect the relevance of polarization cues in vision tasks. Furthermore, Section 3.1 (line 150) will be updated with a subsection on Interpretable Polarization Features, formally defining AoLP, DoP, and CoP, and explaining their visual interpretations, with appropriate citations, including Baek et al. (SIGGRAPH Asia 2018).

---

> > ### Comment · Reviewer_awtX · 2025-08-04
> > **Comment**
> >
> > Thank you for the detailed rebuttals. Please revise the final version to incorporate all the details, visualizations, and related works.

---

> > > ### Author Response · Authors · 2025-08-07
> > >
> > > Thank you very much for your thoughtful comments and positive feedback. We will carefully revise the final version to incorporate all suggested improvements as recommended. We sincerely appreciate your constructive suggestions.

---

### Official Review · Reviewer_ZBuv · 2025-06-30

**Rating:** 6
**Confidence:** 4

**Summary:**

This paper proposes a new task: RGB-to-polarization image estimation, which estimates polarization information from RGB images. Using one of the largest available datasets, the paper benchmarks this task with both restoration-oriented models (e.g., Restormer, Uformer) and generative architectures (e.g., RealFill, Img2ImgTurbo). Through comprehensive experiments, the study offers insights into the strengths and limitations of these approaches, along with effective training strategies. In addition, the paper provides an in-depth discussion on future research directions, outlining potential architectural designs and open challenges.

**Additional Feedback:**

Please refer to the weaknesses section for the major concerns.

Additional suggestions:

1. I suggest the authors include a discussion on potential benchmark sub-tasks under varying optical conditions and propose directions for future dataset enhancements to improve physical diversity and annotation completeness.

2. The parameters of different models should be provided.

**Dataset Code Accessibility:**

Yes

**Dataset Code Comments:**

All datasets used are publicly available, and the implementation code is provided in the supplementary material.

**Ethical Considerations:**

No, there are no or only very minor ethics concerns

**Final Justification:**

The authors have adequately addressed my concerns, and I find that the concerns raised by other reviewers have also been well resolved. Therefore, I am inclined to raise my score.

**Limitations Weaknesses:**

1. The proposed benchmark evaluates numerical metrics (e.g., PSNR, SSIM, and LPIPS) across estimated Stokes components. However, the relationship between these metrics and the Stokes components is not fully explained. A more detailed discussion would strengthen the paper.

2. Could there be multiple polarization states that yield the same RGB appearance? If so, what physical information does this imply? The authors should comment on this in the paper.

3. Polarization arises from various physical processes depending on material properties, such as reflection, scattering, and birefringence. The authors should discuss how different materials give rise to these distinct polarization phenomena and their implications for estimation.

4. Although this paper provides a comprehensive analysis across different models and training strategies, it lacks a discussion on the differences between Stokes components. I found that the estimation performance of different Stokes components varies. Why is that? A more detailed explanation would strengthen the paper.

**Strengths Contributions:**

1. Novel Task Setting: RGB-to-polarization image estimation is a new and meaningful task. Unlike conventional methods that depend on specialized optical hardware, this work introduces a learning-based alternative, enabling broader access to polarization information and expanding its applicability in vision tasks.

2. Solid Benchmark Design: The paper builds a strong benchmark using a large-scale RGB-polarization dataset and evaluates a wide range of models, including both restoration-based and generative architectures. It also compares models trained from scratch and those adapted from pre-trained backbones, offering diverse and informative perspectives.

3. Comprehensive Analysis: Detailed comparisons are made across model types and training strategies, such as restoration vs. generation and pre-training vs. training from scratch. These analyses highlight key insights and identify promising directions for future research in sensor-free polarization estimation.

---

> ### Author Rebuttal · Authors · 2025-07-30
>
> **W1 Relationship Between Numerical Metrics and Stokes Components**
>
> Many thanks for the suggestion. In our benchmark, PSNR and SSIM are used to evaluate pixel-level fidelity and structural similarity between the predicted and ground-truth Stokes components ($S_1$, $S_2$, $S_3$). LPIPS, on the other hand, provides a perceptual similarity score that captures differences in the spatial structure and appearance of the Stokes maps.
>
> The Stokes components represent physical polarization information, and accurate reconstruction, reflected by high PSNR and SSIM, and low LPIPS, indicates better prediction quality. These metrics offer a quantitative assessment of how well the model captures the spatial and structural details of the polarization cues encoded in $S_1$, $S_2$, and $S_3$. The discussion will be included in the revised manuscript to clarify the motivation for metric selection.
>
> **W2 Mapping from RGB to Stokes parameters**
>
> Thank you for raising this question. It is possible for multiple polarization states to result in the same RGB appearance, particularly under unpolarized or partially polarized lighting conditions. This is because RGB values capture intensity and color information, but not the vectorial nature of light, such as polarization orientation and ellipticity, which are encoded in the full Stokes vector ($S_0$, $S_1$, $S_2$, $S_3$). However, the goal of our paper is to provide a standardized benchmark for evaluating the performance of RGB-to-polarization estimation methods. This ambiguity in RGB-to-Stokes mapping applies equally to all methods under comparison and therefore does not affect the fairness or usefulness of the benchmark itself. The mapping will be discussed in the revised manuscript.
>
> **W3 Materials and Polarization Phenomena**
>
> We fully agree with the reviewer’s observation that different materials exhibit distinct polarization responses, reflecting their underlying optical properties. We will include a corresponding discussion in the revised manuscript. For example, specular reflection on smooth surfaces typically produces linearly polarized light aligned with the plane of incidence; volumetric scattering in rough or turbid media reduces the degree of polarization and disrupts polarization direction; and birefringent materials (e.g., certain biological tissues or crystals) introduce phase delays between orthogonal components, thereby enhancing the $S_3$ response.
>
>
> **W4 Estimation Differences Across Stokes Components**
>
> Many thanks for the advice. The difference in estimation performance across Stokes components reflects their distinct physical meanings and varying relevance to RGB information. In the revised manuscript, we will add a discussion to clarify this point.
>
> Specifically, $S_1$ and $S_2$ represent the differences between horizontal vs. vertical, and +45° vs. −45° linear polarization states, respectively. These components are more directly affected by reflection and surface geometry, and often correlate with observable cues in the RGB domain, such as intensity gradients and specular highlights. In contrast, $S_3$ encodes the degree of circular polarization, which is typically weak or near-zero in most natural scenes. Circular polarization has limited influence on visible color or intensity and thus lacks strong visual cues in RGB inputs.
>
> **Q1 Discussion on Potential Benchmark Sub-tasks**
>
> Following the reviewer’s suggestions, in our future work, we plan to consider dividing the benchmark into sub-tasks based on optical conditions, such as varying incident and viewing angles, scenes combining specular and diffuse reflections, and scenes with well-defined material properties (e.g., known refractive index and surface roughness).
>
> In terms of dataset extensions, possible improvements include incorporating geometric information (e.g., surface normals), illumination metadata, material category labels, and high-fidelity physics-based simulation data. These additions will contribute to building a more physically grounded framework for polarization estimation. We will include this discussion in the revised manuscript.
>
>
> **Q2 Parameters of Models**
>
> Many thanks for the feedback. The FLOPs (or MACs), number of parameters, runtime, and memory usage of different methods are summarized below. The runtime is measured on a single NVIDIA RTX A5000 GPU.
> | Method     | FLOPs     | Trainable Parameters | Running Time | Training Memory   | Test Memory |
> |------------|-----------|----------------------|---------------|-------------------|-------------|
> | Restormer  | 38 GFLOPs | 26.1M                | 0.04s         | 2 × 24G GPUs      | ~1G         |
> | Uformer    | 11 GFLOPs | 5.3M                 | 0.01s         | 2 × 24G GPUs      | ~1G         |
> | Wdiff      | 16 GFLOPs | 109.7M               | 0.25s         | 4 × 24G GPUs      | ~1G         |
> | DiT        | 19 GFLOPs | 108.7M               | 0.19s         | 4 × 24G GPUs      | ~1G         |
> | Realfill   | 86 GFLOPs | 1.7M                | 5.00s         | 4 × 24G GPUs      | ~3G         |
> | ImgTurbo   | 86 GFLOPs | 9.5M                | 0.20s         | 4 × 24G GPUs      | ~6G         |
> | MAE    | 64 GFLOPs | 330.0M                 | 0.01s         | 4 × 24G GPUs      | ~2G         |
>
> These metrics will be included in the revised version.

---

> > ### Comment · Reviewer_ZBuv · 2025-08-06
> >
> > The authors have adequately addressed my concerns, and I find that the concerns raised by other reviewers have also been well resolved. Therefore, I am inclined to raise my score.

---

> > > ### Author Response · Authors · 2025-08-07
> > >
> > > Thank you very much for your thoughtful review and for considering raising your score. We sincerely appreciate your constructive feedback.

---

### Comment · Area_Chair_mpQH · 2025-08-03
**Final ratings and justifications.**

Dear Reviews:

This paper has diverse ratings (1 accept, 1 borderline accept, and 1 borderline reject). Now the authors posted responses, can you please check and  provide your final ratings and **very important** justification of your final rating? Thanks.

I noticed one reviewer already started the discussion with the authors (thank you!). Can other reviewers also help to check the rebuttal?

Thanks.

---

### Note · Authors · 2025-08-12

Dear AC and Reviewers,


We sincerely appreciate the time and thoughtful feedback from all reviewers and the AC. We are pleased that three reviewers recognized the contributions of our work, and we are grateful for the supportive suggestions and constructive comments to further improve the paper.
For the final version, we are committed to incorporating all suggested revisions, including:

(a) adding experimental results on a wider range of datasets;

(b) revising the Related Work section to appropriately credit prior contributions;

(c) enhancing visualizations with physical characteristics such as DoP, AoLP and CoP;

(d) adding further evaluations, including downstream task performance, physical constraints, perceptual quality assessments, efficiency analysis, and stability at higher resolutions;

(e) incorporating all suggested content refinements and technical detail enhancements.

Once again, we thank the AC and reviewers for their support, and we hope these improvements will be taken into account when making the final decision.


Sincerely,

Authors

---

### Decision · Program_Chairs · 2025-09-18

**Decision:**

Accept (poster)

**Comment:**

This paper makes a clear and useful contribution by establishing the first benchmark for RGB-to-polarization estimation. While methodological novelty is limited, the benchmark task, dataset aggregation, and systematic evaluation are valuable for the community. The rebuttal strengthened the work significantly by broadening evaluation, adding downstream tasks, and clarifying prior work positioning.
The final reviewer ratings are mixed (but overall positive): one strong accept and two borderline accepts. Supportive reviewers emphasize the benchmark’s potential impact, while borderline reviewers acknowledge solid execution but remain cautious about limited novelty. Nevertheless, the strong accept plus the extensive rebuttal improvements tip the balance toward acceptance. Please revise the final version to incorporate all the details, visualizations, and related works discussed in the rebuttal phase.